# Discrepancy-aware Score Learning for Diffusion Training

## Abstract

Diffusion models excel in stable training and distribution coverage, achieving remarkable results in various generative tasks. However, especially in high-resolution or structurally complex settings, their reliance on denoising score matching (DSM) leads to overly smoothed textures and limited perceptual detail. This limitation arises from DSM's propensity to reduce average reconstruction error rather than addressing challenging perceptual features in the data distribution. We propose Discrepancy-aware Score Learning (DSL), a novel adversarial training framework that incorporates a margin-based energy regularizer to score matching in order to address this challenge. In the noise space, DSL introduces an energy-based discriminator that adaptively highlights samples with high generation discrepancies. Our approach retains the denoising formulation while guiding the generator to prioritize difficult cases. We theoretically connect DSL to Wasserstein gradient flows, interpreting it as functional gradient descent regularized by the discriminator's energy surface. Moreover, we demonstrate that DSL is compatible with the underlying probabilistic model by establishing an equilibrium consistent with the true score function. Compared to baseline diffusion models and recent adversarial approaches, DSL significantly improves sample fidelity, perceptual sharpness, and semantic alignment, according to extensive experiments conducted across text-to-image generation, conditional synthesis, super-resolution, and 2D-to-3D reconstruction.

## 1 Introduction

Diffusion models have emerged as a foundational technique in generative modeling, achieving state-of-the-art results on a wide range of tasks, including image synthesis Ho et al. (2020); Rombach et al. (2022), video generation Singer et al. (2023) and 3D reconstruction Li et al. (2024a); Liu et al. (2023b); Long et al. (2024). These models learn to reverse a Markovian noising process by estimating the score (*i.e.*, the log density gradient) of the data distribution at each step Song et al. (2021b). Thanks to their probabilistic grounding and robust training behavior, diffusion models provide superior distribution coverage and sample diversity in comparison to generative adversarial networks (GANs) Goodfellow et al. (2014).

Despite their empirical success, diffusion models often encounter challenges related to perceptual quality due to their dependence on denoising score matching (DSM) Ho et al. (2020). This approach aims to minimize the mean squared error between the predicted noise and the actual noise. This formulation leads to blurred textures and limited high-frequency detail, especially in high-resolution contexts Dhariwal & Nichol (2021); Vahdat et al. (2021). The underlying issue is that DSM encourages the minimization of the average error across the entire distribution, rather than prioritizing perceptually significant modes. Recent research has explored integrating diffusion training with adversarial objectives Wang et al. (2023); Salimans et al. (2022); Nie et al. (2022). This leverages GANs Goodfellow et al. (2014) to enhance sharpness and improve perceptual realism. However, these methods often depend on discriminators that are trained independently in either image space or latent space. This approach may introduce potential instability, as it can conflict with the likelihood-based training objectives inherent to the diffusion model.

To address these limitations, we propose Discrepancy-aware Score Learning (DSL), an adversarial training framework that extends the score-matching paradigm with energy-based regularization. Unlike previous adversarial diffusion methods using separate discriminators in pixel or latent space, DSL employs an energy-based discriminator that differentiates between real and generated noise

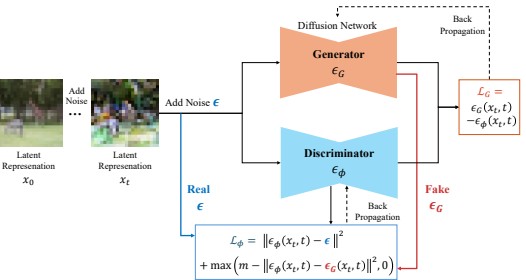

Figure 1: High-level overview of the proposed DSL framework. The generator is optimized via denoising score matching to estimate $\epsilon_G$, and the discriminator is optimized via a margin-based adversarial energy loss to separate $\epsilon_G$ from true noise $\epsilon$.

with a margin-based hinge loss Zhao et al. (2017). This discriminator guides the generator in producing sharper, consistent noise estimates, functioning as a learnable reweighting mechanism in the score matching process. In particular, DSL preserves the foundational structure of diffusion models, in which the generator functions as a noise-predicting denoiser. The introduced energy-based hinge loss adaptively reweights gradients based on sample discrepancies, thereby encouraging the model to focus on challenging regions and enabling self-correcting regularization. We interpret DSL training as a functional gradient descent in a distribution space regularized by the discriminator's energy surface. This perspective connects DSL with Wasserstein gradient flows Ambrosio et al. (2008); Zhang et al. (2020), offering theoretical insights into its convergence behavior and informing design choices. Moreover, we show that DSL admits a well-defined equilibrium consistent with the true score function, even in the presence of nonzero adversarial margins, thus formally guaranteeing compatibility with conventional score matching.

Our framework is motivated by the observation that conventional diffusion training lacks a mechanism for adaptively focusing learning in areas where the generator exhibits suboptimal performance. By employing an adversarial discriminator that is trained to penalize suboptimal generations beyond a specified margin, DSL acts as a self-correcting regularizer that dynamically re-weights gradients throughout the training process. This improves robustness in low-data regimes, improves generalization across noise schedules, and enhances perceptual quality without reliance on pre-trained perceptual networks Zhang et al. (2018). To validate the versatility and generalizability of our method, we conduct comprehensive experiments across a range of generative tasks. Our evaluations include text-to-image synthesis, conditional generation Li et al. (2023); Gal et al. (2022a), image super-resolution Wang et al. (2024a), and 2D-to-3D generation Liu et al. (2023b); Li et al. (2024a). We demonstrate consistent improvements in perceptual quality, semantic alignment, and geometric coherence across all benchmarks.

In summary, our main contributions are as follows: (1) We introduce *Discrepancy-aware Score Learning (DSL)*, a novel adversarial training framework for diffusion models that enhances score learning through an energy-based hinge loss, without requiring architectural changes to the standard denoising pipeline; (2) We present a *functional gradient interpretation* of DSL in the space of distributions, connecting its training dynamics to the Wasserstein gradient flows; (3) We formulate DSL as a *margin-based energy minimization* approach that generalizes denoising score matching while maintaining compatibility with the learned generative prior; and (4) Extensive experiments across diverse generation tasks, including text-to-image synthesis, conditional generation, image super-resolution, and 2D-to-3D reconstruction, demonstrate that DSL consistently improves sample fidelity, perceptual realism, and semantic alignment compared to existing baselines.

## 2 DISCREPANCY-AWARE SCORE LEARNING

This section presents *Discrepancy-aware Score Learning (DSL)*, a framework that merges diffusion-based denoising score matching with a margin-based adversarial discriminator Zhao et al. (2017) to enhance stability and convergence in the reverse diffusion process Ho et al. (2020); Song et al. (2021b). Figure 1 provides high-level overview of DSL. The generator $\epsilon_G(x_t, t)$ is trained via denoising score matching to predict the noise $\epsilon_G$ introduced during diffusion. The discriminator network $\epsilon_\phi(x_t, t)$ employs a margin-based adversarial energy loss to differentiate predicted noise $\epsilon_G$ from true noise $\epsilon$. Training is thus driven by two complementary objectives: the score matching loss $\mathcal{L}_G$ for the generator and adversarial energy loss $\mathcal{L}_\phi$ for the discriminator.

## 2.1 Adversarial Objective with Margin-based Energy

Let $x_0 \sim p(x_0)$ be a clean data sample and $\epsilon \sim \mathcal{N}(0, \mathbf{I})$ be the Gaussian noise added during the forward diffusion process. At time step $t$, the noisy sample is:

$$x_t = \alpha_t x_0 + \sigma_t \epsilon, \tag{1}$$

where $\alpha_t, \sigma_t$ are predefined schedule coefficients.

The generator network $\epsilon_G(x_t, t)$ predicts noise added to $x_t$, while the discriminator network $\epsilon_\phi(x_t, t)$ evaluates the plausibility of a noise vector, functioning as an energy-based model. Following the EB-GAN formulation Zhao et al. (2017), we use a hinge-style margin loss for the discriminator. The discriminator is trained to assign low energy to real noise samples $\epsilon$, and enforce a margin between its output and the generator's prediction:

$$\mathcal{L}_\phi = \mathbb{E}_{x_t, t} \left[ \|\epsilon_\phi(x_t, t) - \epsilon\|^2 + \lambda \cdot \max \left( 0, m - \|\epsilon_\phi(x_t, t) - \epsilon_G(x_t, t)\|^2 \right) \right], \tag{2}$$

where $\lambda$ is a weighting factor and $m$ is a positive margin. The generator learns to minimize the distance between its predictions and the discriminator's output, producing noise vectors indistinguishable from real ones:

$$\mathcal{L}_G = \mathbb{E}_{x_t, t} \left[ \|\epsilon_G(x_t, t) - \epsilon_\phi(x_t, t)\|^2 \right]. \tag{3}$$

This adversarial approach establishes a self-corrective training loop where the generator is guided by the discriminator's dynamically updated scoring function. Subsequent sections demonstrate that this design enhances alignment with the true noise distribution and mitigates instability and overfitting commonly seen in direct score regression.

## 2.2 Functional Gradient Interpretation

Discrepancy-aware Score Learning (DSL) can be interpreted as a functional gradient descent in the space of probability distributions over noise. It connects DSL to the broader framework of variational inference and Wasserstein gradient flows Ambrosio et al. (2008); Zhang et al. (2020), providing theoretical insight into how generator updates converge to the true score function.

Let $q_G(\epsilon|x_t)$ denote the implicit distribution over noise induced by the generator $\epsilon_G$, and let $p(\epsilon|x_t)$ be the true noise distribution, typically $\mathcal{N}(0, \mathbf{I})$ due to the forward diffusion process. The generator's training goal is minimizing the KL divergence:

$$\min_{\epsilon_G} \mathbb{E}_{x_t} \left[ D_{\mathrm{KL}}(q_G(\epsilon|x_t) \| p(\epsilon|x_t)) \right], \tag{4}$$

which, assuming Gaussian distribution, relates to the expected squared error:

$$D_{\mathrm{KL}}(q_G \| p) \propto \mathbb{E}_{x_t} \left[ \|\epsilon_G(x_t, t) - \epsilon\|^2 \right]. \tag{5}$$

In DSL, rather than accessing $\epsilon$ directly, the generator minimizes its distance to the dynamically learned score estimate $\epsilon_\phi$, which is trained to approximate $\epsilon$ while actively rejecting incorrect predictions via the margin-based constraint. As shown in Eq. (3), the generator optimizes its output toward $\epsilon_\phi$, effectively following a learned gradient flow shaped by the discriminator. This formulation allows DSL to flexibly guide the generator through a denoised energy landscape, rather than relying solely on noisy supervision from $\epsilon$.

From the Wasserstein gradient flow perspective Ambrosio et al. (2008), the update of $q_G$ follows the continuity equation:

$$\frac{dq_G}{d\tau} = -\nabla \cdot \left( q_G \nabla \log \frac{q_G}{p} \right), \tag{6}$$

where $\tau$ is a virtual gradient flow time, and the descent direction is determined by KL divergence. Our formulation incorporates an energy-based loss landscape from $\epsilon_\phi$, leading to a more stable and interpretable evolution of the generator's score function. Thus, DSL is a regularized gradient flow that refines the generator's noise predictions toward the true score function, utilizing the discriminator as a proxy for the gradient direction. This framework generalizes and stabilizes conventional score-matching learning.

## 2.3 Margin-aware Convergence Analysis

We extend the equilibrium analysis of DSL into the positive-margin regime. The discriminator and generator objectives are defined in (2) and (3). For $m = 0$, the system simplifies to conventional denoising score matching; when $m > 0$, the hinge term introduces an energy cutoff akin to the EB-GAN formulation Zhao et al. (2017). The following theorem demonstrates that equilibrium persists and quantifies the maximum deviation from the true noise under the margin constraint.

**Theorem 1** (Margin-aware equilibrium). *Assume the generator $\epsilon_G$ and discriminator $\epsilon_\phi$ have sufficient capacity. Let $(\epsilon_G^\star, \epsilon_\phi^\star)$ jointly minimize Eq. (3) and maximize Eq. (2) for a fixed margin $m > 0$. Then, for every $(x_t, t)$,*

$$\epsilon_\phi^\star(x_t, t) = \epsilon, \qquad \|\epsilon_G^\star(x_t, t) - \epsilon\|^2 \le m. \tag{7}$$

*Thus, $\epsilon_G^\star$ converges to the closed $m$-ball $\mathcal{B}_m(\epsilon) = \{\hat{\epsilon} \mid \|\hat{\epsilon} - \epsilon\|^2 \le m\}$ centered at the true noise.*

*Proof.* Because $\epsilon$ is available to the discriminator, Eq. (2) is minimized when $\epsilon_\phi(x_t, t) = \epsilon$, independently of $m$. Substituting this choice into Eq. (3) yields

$$\mathcal{L}_G = \|\epsilon_G(x_t, t) - \epsilon\|^2, \tag{8}$$

subject to the implicit constraint imposed by the hinge penalty, $\|\epsilon_G(x_t, t) - \epsilon\|^2 \ge m$. The optimum is therefore attained at the boundary $\|\epsilon_G - \epsilon\|^2 = m$ if $m > 0$, and at $\epsilon_G = \epsilon$ when $m = 0$. Hence the stated bound holds and the set $\mathcal{B}_m(\epsilon)$ is the equilibrium neighbourhood. $\square$

Theorem 1 characterizes the equilibrium behaviour of DSL under a positive margin. While $m = 0$ yields exact score matching, the margin-based setting $m > 0$ provably constrains the generator within an $m$-radius ball around the ground-truth noise. Crucially, this structure not only preserves stability but also it actively improves the learning dynamics through two mechanisms that enhance convergence and generalization.

First, the margin induces *gradient amplification* on high-error samples. From Eq. (2), whenever the generator prediction falls within the hinge region, the generator receives an additional directional gradient toward the discriminator's estimate:

$$\nabla_{\epsilon_G} \left[ \lambda \max(0, m - E) \right] = -2\lambda(\epsilon_\phi - \epsilon_G) \cdot \mathbf{1}_{E < m}. \tag{9}$$

This term selectively increases the gradient norm for samples violating the margin constraint. Consequently, training prioritizes correcting large errors early, resulting in faster convergence and improved alignment of $\epsilon_G$ to $\epsilon$ in fewer iterations.

Second, the discriminator serves as a *variance-reducing proxy* for true noise. Since $\epsilon_\phi$ is trained to approximate $\epsilon$ while rejecting erroneous outputs, it provides a denoised supervision signal. Let $\epsilon_\phi = \epsilon + \nu$ where $\mathbb{E}[\nu] = 0$ and $\mathrm{Var}(\nu) < \mathrm{Var}(\epsilon)$. Thus, the generator's regression target in Eq. (3) has lower variance compared to standard DSM:

$$\mathrm{Var}(\epsilon_G - \epsilon_\phi) < \mathrm{Var}(\epsilon_G - \epsilon). \tag{10}$$

This mitigates overfitting to noisy supervision and enhances generalization performance, particularly under limited data or high-dimensional noise. Overall, the positive margin strengthens DSL theoretically and practically: it facilitates targeted error correction through hinge-induced gradient shaping and lowers estimator variance via learned denoising supervision. These effects lead to tighter convergence around true noise while enhancing generalization bounds, making margin-aware training both stable and fundamentally beneficial.

### 2.4 CONNECTION TO SCORE MATCHING AND GENERALIZATION

DSL can be viewed as a strict generalization of the conventional denoising score matching (DSM) objective widely used in diffusion models Ho et al. (2020); Song et al. (2021b). In this section, we show that DSL reduces to DSM under specific conditions and discuss how its adversarial margin formulation provides additional structure for more stable training.

**Reduction to DSM.** Let us consider the case when the discriminator is replaced by the ground-truth noise vector $\epsilon$, *i.e.*, $\epsilon_\phi(x_t, t) = \epsilon$. Substituting into the generator loss in (3), we obtain:

$$\mathcal{L}_G = \mathbb{E}_{x_t, t} \left[ \|\epsilon_G(x_t, t) - \epsilon\|^2 \right] = \mathcal{L}_{\mathrm{DSM}}. \tag{11}$$

Alternatively, when the margin parameter $m \to 0$, the discriminator loss in (2) simplifies to a regression loss, and the margin-aware analysis in Sec. 2.3 guarantees that both networks converge to the ground-truth noise in the limit as $m \to 0$.

*Generalization and Training Benefits.* Beyond this special case, DSL provides several practical benefits during training. First, when the generator is far from producing realistic noise, the hinge loss component in the discriminator amplifies the corrective gradient signal, especially in early stages of learning. This facilitates faster and more stable convergence. Second, the use of a learned discriminator acts as a smoothing filter that replaces noisy supervision with a cleaner, dynamically

Table 1: Evaluation of performance comparisons with SD XL Podell et al. (2024).

| Method | FID ($\downarrow$) | CLIP score ($\uparrow$) |
|---|---|---|
| Stable Diffusion | 13.68 | 0.3102 |
| DSL (Ours) | **9.21** | **0.3481** |

shaped energy surface. This regularization helps mitigate overfitting and improves generalization, especially under noisy data or high-variance sampling conditions. Finally, the margin term introduces an explicit separation between real and fake noise, effectively encouraging the generator to avoid collapsing into ambiguous or non-informative solutions. This behavior aligns with findings in margin-based adversarial learning Gulrajani et al. (2017), which show that enforcing geometric separation stabilizes optimization and leads to more interpretable model behavior. In summary, DSL extends the score-matching paradigm with adversarial structure and energy-based regularization, providing both theoretical consistency and practical improvements to training dynamics.

## 3 EXPERIMENTS

In this section, we conducted experiments to evaluate the performance of the proposed method across diverse generative tasks, including the standard text-to-image generation Podell et al. (2024) (Sec. 3.2.1), conditional text-to-image generation Li et al. (2023); Gal et al. (2022a) (Sec. 3.2.2), image super-resolution Wang et al. (2024a) (Sec. 3.2.3), and 2D-to-3D generation Liu et al. (2023b); Li et al. (2024a) (Sec. 3.2.4).

### 3.1 EXPERIMENTAL DETAILS

To ensure fair comparisons, both the baseline method and comparative experiments for each generation task were trained with the same number of steps and identical parameter configurations. In particular, we used the SDXL Podell et al. (2024); Rombach et al. (2022) as the base diffusion architecture across all experimental settings. For computational efficiency, all experiments in this section were conducted by fine-tuning pre-trained models. Additionally, results from training the same architecture from scratch are reported in Sec. B of the *supplemental material*.

We set $\lambda = 0.05$ in all experiments, yielding optimal results in the sensitivity analysis. The sensitivity analysis of the weighting factor $\lambda$ is in Sec. C of the *supplemental material*. We computed the mean error to set the margin $m$, conducting inference on the training dataset every five epochs. This follows the decision-theoretic principle that the conditional mean is the optimal estimate under a squared error loss function Lehmann & Casella (1998). This methodology aligns with previous research in energy-aware adversarial optimization Zhao et al. (2017); Grathwohl et al. (2020), allowing the margin to be adaptively adjusted to the data distribution and mitigating the risks of overestimation and underestimation. Varying the margin value within a $\pm 10\%$ did not yield statistically significant performance differences. This outcome indicates that the adaptive margin strategy maintains a significant separation between true and generated latent representations during training and remains robust to moderate variations.

In our experiments, we implemented the discriminator using Low-Rank Adaptation (LoRA) Hu et al. (2022). The discriminator reuses a frozen copy of the generator U-Net backbone, injecting rank-8 learnable low-rank matrices (*i.e.*, LoRA adapters) into the linear projection layers of the attention modules. This design incurs minimal memory and computational overhead, allowing to maintain the same batch size and hardware configuration as the baseline. Refer to Sec. B for the detailed analysis of memory and computational overhead introduced by the discriminator.

### 3.2 PERFORMANCE COMPARISONS

#### 3.2.1 TEXT-TO-IMAGE GENERATION.

To evaluate the proposed method, we conducted a comparative experiment with the baseline model, *i.e.*, Stable Diffusion Podell et al. (2024); Rombach et al. (2022). To ensure fair comparison, all models were trained on the LAION-5B dataset Schuhmann et al. (2022). For evaluation, we sampled 20K image-text pairs from the COCO2014 validation set Lin et al. (2014) and the Flickr30k dataset Young et al. (2014). Following previous work Zhang et al. (2023); Balaji et al. (2022), we adopted two commonly used metrics: the Fréchet Inception Distance (FID) Heusel et al. (2017) to evaluate the distributional similarity between real images and generated images, and the Contrastive Language-Image Pre-training (CLIP) score Radford et al. (2021), a normalized metric for assessing the semantic match between the text prompt and the generated image using CLIP embeddings.

**Results with Stable Diffusion.** Table 1 compares the performance of Stable Diffusion with and without the proposed method, using FID and CLIP score. The results clearly show that the proposed

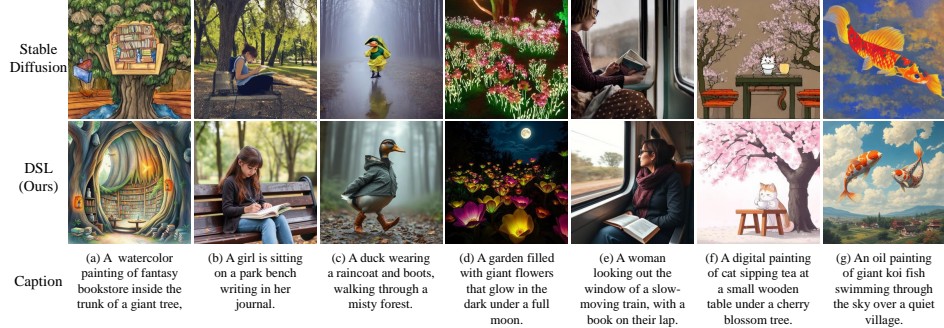

Caption

(a) A watercolor painting of fantasy bookstore inside the trunk of a giant tree,

(b) A girl is sitting on a park bench writing in her journal.

(c) A duck wearing a raincoat and boots, walking through a misty forest.

(d) A garden filled with giant flowers that glow in the dark under a full moon.

(e) A woman looking out the window of a slow-moving train, with a book on their lap.

(f) A digital painting of cat sipping tea at a small wooden table under a cherry blossom tree.

(g) An oil painting of giant koi fish swimming through the sky over a quiet village.

Figure 2: Sample comparison with the proposed method and Stable Diffusion Podell et al. (2024); Rombach et al. (2022) trained on the LAION-5B dataset Schuhmann et al. (2022).

Table 2: Comparisons with conditional text-to-image methods Li et al. (2023); Gal et al. (2022a).

| Method | FID ($\downarrow$) | CLIP score ($\uparrow$) | CLIP similarity ($\uparrow$) |
|---|---|---|---|
| GLIGEN | 11.83 | 0.279 | - |
| DSL (Ours) | **9.01** | **0.321** | - |
| Textual Inversion | 12.74 | - | 0.748 |
| DSL (Ours) | **8.36** | - | **0.797** |

method outperforms the baseline across all metrics. This demonstrates the effectiveness of incorporating an energy-based discriminator during the diffusion process, allowing self-correcting regularization. Consequently, our method improves both semantic alignment and distributional similarity. The qualitative results are illustrated in Fig. 2. In Fig. 2 (a), our method depicts a detailed "fantasy bookstore inside the trunk of a giant tree", while the baseline lacks structural clarity, producing a less plausible result. Similar advantages are seen in human depiction and complex scene understanding. In Figs. 2 (b) and (e), the proposed method captures accurate body posture and facial expressions for prompts, whereas the baseline struggles with anatomical coherence and expressiveness. In general, both quantitative and qualitative comparisons confirm the fidelity, semantic expressiveness, and structural accuracy achieved by our method, especially under challenging generative conditions.

### 3.2.2 CONDITIONAL TEXT-TO-IMAGE GENERATION.

We evaluated the effectiveness and extensibility of our method on two conditional text-to-image generation frameworks: GLIGEN Li et al. (2023) and Textual Inversion Gal et al. (2022a). We compared the performance of these methods with and without our approach. GLIGEN is an extension of the text-to-image diffusion model that incorporates a localization layer for image generation using conditions such as bounding boxes, keypoints, and semantic maps. We assessed the proposed method's performance through experiments with GLIGEN, using bounding boxes and captions as conditions. Both variants trained on SBU Ordonez et al. (2011) and CC3M Changpinyo et al. (2021), using bounding boxes from GLIP Li et al. (2022). We sampled 2K image-text pairs from the Flickr30k dataset for evaluation Young et al. (2014). We used the FID Heusel et al. (2017) for distributional similarity and the CLIP score Radford et al. (2021) for semantic alignment between images and their descriptions. Textual Inversion facilitates the learning of novel visual concepts within the text embedding space of diffusion models, utilizing minimal input images. This process enables the generation of personalized images. We evaluated the performance of the proposed method using a limited dataset via comparative experiments. The Google Scanned Objects (GSO) Downs et al. (2022) dataset and a custom multi-view dataset were used, selecting six $512{\times}512$ images per concept. We measured performance using the FID and the CLIP similarity Gal et al. (2022b), which is defined as the average pairwise cosine similarity between the CLIP image embeddings of the input and generated images. This assessment evaluates visual consistency in few-shot conditions.

**Results with GLIGEN.** Table 2 compares performance results to GLIGEN Li et al. (2023), evaluated using the FID and the CLIP score. The results shows that integrating our method with GLIGEN the baseline improves both image fidelity and conditional alignment, leading to more accurate bounding box positioning and visually coherent object placement. Qualitative comparisons are illustrated in Fig. 3. The results show that our method generates images that better respect the conditional layout while maintaining semantic consistency and high visual fidelity. In the first example, the baseline method introduces a duplicated "stick" and renders the dog in an awkward pose, failing to convey the natural motion of "swimming". In contrast, our method accurately aligns the bounding boxes of all entities and generates a coherent scene that faithfully captures the intended action and spatial relationships. These advantages become clearer with increasing scene complexity. For

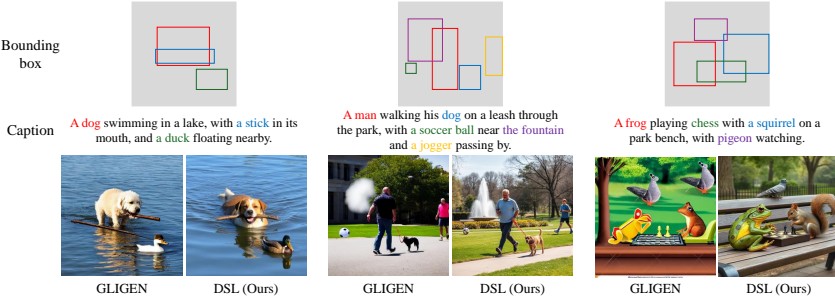

Figure 3: Comparisons with GLIGEN Li et al. (2023) conditioned on a bounding box. Both methods are trained on SBU Ordonez et al. (2011) and CC3M Changpinyo et al. (2021) datasets.

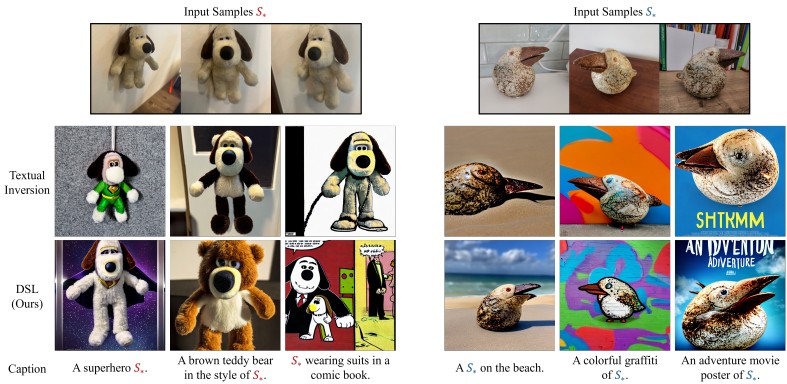

Figure 4: Comparisons with Textual Inversion Gal et al. (2022a). We trained with 6 input images per object. $S_*$ is a pseudo-word representing the embedding vector for the input concept.

instance, in the second example, the proposed method places all objects such as "the fountain" and "a soccer ball" more accurately and naturally than the baseline, while maintaining harmony among the entities. This demonstrates that our model is not only effective at aligning spatial constraints but also excels at integrating fine-grained visual details in a realistic manner.

**Results with Textual Inversion.** Table 2 compares the performance of Textual Inversion with and without the proposed method, using FID and CLIP similarity. The results show that incorporating DSL significantly improves both metrics. In particular, lower FID indicates that adversarial training enhances robustness in low-data regimes and better captures data distribution, resulting in more realistic, high-fidelity images. Also, higher CLIP similarity highlights that the proposed method effectively preserves visual consistency and semantic identity.

Figure 4 provides qualitative comparisons between Textual Inversion and DSL. The results show that our method accurately preserves the characteristics of the input samples while generating diverse images that align well with the conditioning text. For instance, in the first and second row of first sample in Fig. 4 involve changes in appearance and geometry. While the baseline often overfits to training samples and fails to reflect textual variation, our method preserves object identity and flexibly adapts to various scenes, producing consistent yet diverse generations. Furthermore, as shown as in the final row of the first and second sample, the superiority of DSL is especially evident in artistic and design styles such as comic books and posters. These results confirm the robustness and generalizability of the proposed method in accurately capturing object-specific concepts while adapting to various textual prompts and styles.

### 3.2.3 IMAGE SUPER-RESOLUTION.

To evaluate the effectiveness and extensibility of the proposed method, we conducted comparative experiments with StableSR Wang et al. (2024a), a recent diffusion-based image super-resolution model. For a fair comparison, both models were trained on the AID dataset Xia et al. (2017). We randomly selected 100 images from each of the 20 categories, totaling 2K samples, with each image resized to 512×512 resolution. For evaluation, ten non-overlapping images from each category were selected, totaling 200 images. Each test image was downsampled to 128×128 before being processed by the super-resolution models. We assessed super-resolution performance using FID, Peak Signal-to-Noise Ratio (PSNR) Wang et al. (2004), and Learned Perceptual Image Patch Similarity (LPIPS) Zhang et al. (2018), following recent literature Saharia et al. (2022); Luo et al. (2023).

Table 3: Comparisons with StableSR Wang et al. (2024a) trained on AID dataset Xia et al. (2017).

| Method | FID (↓) | PSNR (↑) | LPIPS (↓) |
|---|---|---|---|
| StableSR | 25.72 | 23.67 | 0.3216 |
| DSL (Ours) | **24.13** | **25.85** | **0.3083** |

| Zoomed LR | StableSR | DSL (Ours) | Zoomed LR | StableSR | DSL (Ours) |

Figure 5: Sample comparisons with the proposed method and StableSR Gal et al. (2022a).

**Results with StableSR** The results of the comparison with StableSR are presented in Table 3. The results show that the proposed method consistently outperforms the baseline across all evaluation metrics. In particular, the reduction in FID and the increase in PSNR highlight the effectiveness of our proposed structural consistency energy, which leverages adversarial training to suppress blurring artifacts common in MSE-based diffusion and enhance high-resolution structural learning. This performance advantage is further validated by the qualitative comparison in Fig. 5. In the first sample, our method clearly delineates textures such as grasses and road lane markings, while the baseline produces blurred boundaries and homogenized textures across the field, road, and lawn. The second sample shows a similar trend, with architectural elements and road surfaces appearing sharper and more faithful to the ground truth in our result compared to StableSR. Overall, our approach demonstrates enhanced effectiveness and generalizability in preserving spatial structures and surface detail, reaffirming its strength in high-fidelity super-resolution tasks.

### 3.2.4 2D-TO-3D GENERATION.

To evaluate the effectiveness and extensibility of the proposed method, we compared recent 2D-to-3D methods Liu et al. (2023b); Li et al. (2024a) with and without DSL. Specifically, we used SyncDreamer Liu et al. (2023b) and Era3D Li et al. (2024a) as baselines. SyncDreamer produces consistent multi-view images from a single-view input by using 3D-aware feature attention to synchronize intermediate states across views, ensuring geometric and color consistency. In contrast, Era3D synthesizes multi-view images from a single-view input by predicting camera parameters and integrating cross-view information, utilizing row-wise attention to enforce epipolar priors. All models were trained using renderings from a subset of the Objaverse dataset Deitke et al. (2023), with each image at $512 \times 512$ resolution.

To evaluate SyncDreamer with and without DSL, we sampled 32 views by generating 16 views with uniformly distributed azimuth angles at an elevation of $30°$. Additionally, we generated 16 views with the same azimuths but with random elevations from the range $[-20°, 40°]$. For Era3D, the training utilized both orthographic and perspective views. Specifically, we first rendered 16 images with an orthogonal camera at sampled azimuths and a fixed elevation of $0°$. For each azimuth, we rendered three images with perspective cameras and one with an orthogonal camera, using random elevations from the range $[-20°, 40°]$. We used randomly chosen single-view renderings of 30 objects from GSO Downs et al. (2022) and OmniObject3D (Omni3D) Wu et al. (2023). We also assessed performance using images from the Internet. We evaluated both *novel view synthesis* and *3D reconstruction* to assess the generative performance of the models. For novel view synthesis, we used PSNR, Structural Similarity Index Measure (SSIM) Wang et al. (2004), and LPIPS Zhang et al. (2018). For 3D reconstruction, we followed recent works Liu et al. (2023a); Long et al. (2024) and used Chamfer Distance (CD) and Volume Intersection over Union (IoU) as evaluation metrics.

**Results with SyncDreamer.** Table 4 reports performance comparisons between SyncDreamer and our proposed method across novel view synthesis and 3D reconstruction tasks. The results indicate that incorporating the proposed method consistently outperforms the baselines in both tasks. In particular, the improvements in PSNR, SSIM, and LPIPS is primarily due to our training framework, which integrates energy-based adversarial training to adaptively reweight gradients and allows the model to learn more effectively from challenging regions that standard diffusion training tends to overlook. As a result, the generated multi-view images exhibit improved view consistency and visual sharpness, ultimately leading to more accurate and coherent 3D reconstructions. Figure 6

Table 4: Performance comparisons of multi-view images and textured mesh utilizing recent 2D-to-3D methods Liu et al. (2023b); Li et al. (2024a).

| Method | PSNR (↑) | SSIM (↑) | LPIPS (↓) | CD (↓) | IoU (↑) |
|---|---|---|---|---|---|
| SyncDreamer | 20.17 | 0.756 | 0.136 | 0.0232 | 0.5354 |
| DSL (Ours) | 22.32 | 0.846 | 0.115 | 0.0154 | 0.6749 |
| Era3D | 21.05 | 0.789 | 0.138 | 0.0174 | 0.6513 |
| DSL (Ours) | **23.83** | **0.837** | **0.116** | **0.0125** | **0.7231** |

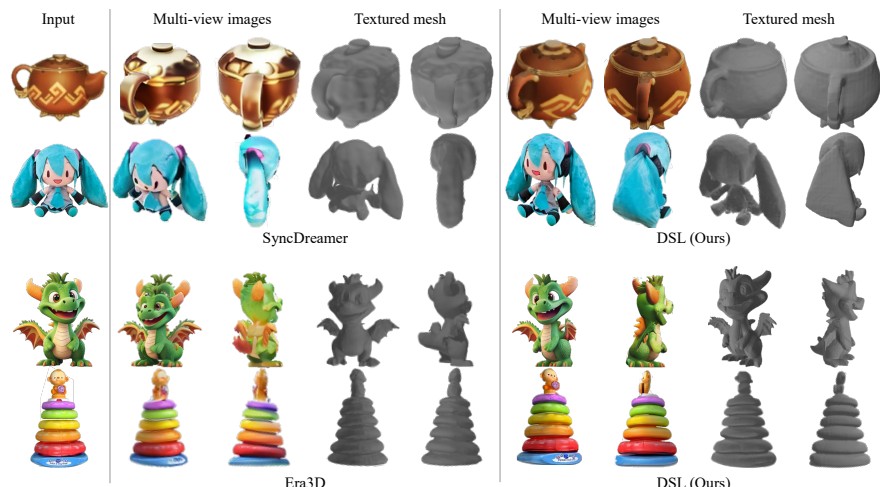

Figure 6: Comparisons of multi-view images and textured mesh generated with SyncDreamer Liu et al. (2023b) and Era3D Li et al. (2024a).

illustrates qualitative comparisons, where our method consistently improves geometric alignment and texture continuity across views, producing sharper and refined textured meshes. In the first sample, our method accurately reconstructs geometric elements (*e.g.*, the teapot handle and base) and fine surface textures (*e.g.*, the side patterns and lid details), in contrast to the baseline, which loses detail. Moreover, as shown in the second sample, our method avoids over-smoothing and minimizes artifacts while preserving geometric structure, texture, and lighting, maintaining inter-view consistency and producing sharp and realistic 3D reconstructions. These results affirm the robustness and scalability of our approach across diverse object categories and visual complexities.

**Results with Era3D** Table 4 summarizes Era3D's performance with and without the proposed method. Results show that the proposed method significantly improves all evaluation metrics over the baseline. Notably, the gains in CD and IoU demonstrate that the energy-based discriminator effectively enhances structural alignment and geometric integrity for Era3D. As a result, the model produces consistent, sharp multi-view images, resulting in detailed, high-quality 3D reconstructions. Improvements are shown in Fig. 6. In the third sample, Era3D oversmooths details such as the dragon's wings and facial expression, causing artifacts. In contrast, our method preserves high-frequency details and sharply reconstructs wings, facial features, claws, and hair. In the fourth sample, the baseline blurs the boundaries between stacked discs, deforming the monkey figure and compromising its pose and expression. In contrast, our method reconstructs each disc with clear separation and captures the monkey's pose and expression consistently across views. These results confirm that our method enhances numerical performance while preserving structural fidelity and semantic detail in complex 3D reconstructions.

## 4 CONCLUSION

In this work, we presented Discrepancy-aware Score Learning (DSL), an adversarial training framework leveraging energy-based regularization to enhance denoising score matching in diffusion models. To guide the generator toward focusing on samples with high estimation errors while maintaining the initial score-matching goal, DSL adds a discriminator to the noise space. We showed that DSL is capable of being acknowledged as functional gradient descent under an energy landscape and theoretically proved that its equilibrium is consistent with the actual score function. DSL continuously enhances sample fidelity, perceptual realism, and semantic consistency without necessitating architectural changes or external perceptual losses, as demonstrated by extensive experiments conducted across text-to-image generation, conditional synthesis, image super-resolution, and 2D-to-3D generation. While our framework targets image-domain unconditional and conditional generation tasks, future research could extend DSL to modalities such as audio or video synthesis.

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
