## A    RELATED WORKS

### A.1    DENOISING SCORE MATCHING AND DIFFUSION MODELS

Denoising Score Matching (DSM) Ho et al. (2020) is key for training modern diffusion models. Originally proposed as a tractable score matching approximation, DSM optimized a neural network to predict noise in corrupted data points. This approach enables iterative denoising through a learned reverse process. Score-based generative modeling Song et al. (2021b) has further generalized this perspective by formulating diffusion models as discretizations of stochastic differential equations (SDEs); noise prediction networks serve as estimators of the score function associated with the data distribution. Advancements have been made on this foundation to enhance generation quality and improve sample efficiency. Denoising Diffusion Probabilistic Models (DDPM) Ho et al. (2020) formalized the DSM framework for Gaussian noise, while Denoising Diffusion Implicit Models (DDIM) Song et al. (2021a) offered a non-Markovian deterministic approach that improves sampling efficiency. Recent works Dhariwal & Nichol (2021); Vahdat et al. (2021) have explored architectural and objective enhancements to improve fidelity and expressiveness. These enhancements include classifier guidance Dhariwal & Nichol (2021) and latent space modeling Vahdat et al. (2021).

Despite their strengths, DSM-based diffusion models often oversmooth textures due to the regression-to-mean effect. This issue is especially evident in high-resolution or perceptually complex tasks. Our research enhances the score estimation framework with a discrepancy-aware energy-based regularizer. This regularizer is designed to guide the learning process toward regions exhibiting elevated error rates while simultaneously preserving the fundamental denoising formulation.

### A.2    ADVERSARIAL TRAINING IN DIFFUSION MODELS

Recent research combines adversarial learning with diffusion models to improve visual fidelity and robustness. Initial approaches, such as Diffusion-GAN Nie et al. (2022) and Progressive Distillation Salimans et al. (2022), employed adversarial loss to enhance texture details and improve sampling efficiency. However, these models rely on externally trained discriminators operating in pixel or latent space, often optimized separately from the score-matching objective, leading to instability and integration challenges. In contrast, our method introduces a noise-space discriminator, architecturally similar to the generator, trained jointly with margin-based regularization. This ensures consistent score prediction and aligns with the original denoising objective.

More recently, advanced adversarial diffusion methods have emerged in prestigious academic venues. Structure-guided Adversarial Training Yang et al. (2024) introduces a batch-level structure-aware discriminator, improving domain generalization and multi-modal synthesis. Adversarial Diffusion Distillation (ADD) Sauer et al. (2024) combines distillation with GAN loss for faster sampling without compromising generation quality. Other works, such as Diffusion Contrastive Purification Wang et al. (2024b), employ contrastive learning to enhance robustness to adversarial attacks. In layout-to-image synthesis, ALDM Li et al. (2024b) integrates adversarial objectives to align scene layouts with synthesized images, thereby improving controllability. Defensive training has also been employed to eliminate concepts from diffusion models under adversarial supervision Fan et al. (2024).

Compared to these approaches, our method, *i.e.*, DSL, introduces a discriminator directly in the noise space, formulating the learning process as margin-based energy minimization. This avoids re-designing the sampling process or integrating external perceptual modules. By aligning the training dynamics with functional gradient descent, DSL provides a theoretically grounded and practically effective framework that generalizes score matching while maintaining architectural simplicity.

### A.3    THEORETICAL PERSPECTIVES ON SCORE MATCHING

Score-based diffusion models are based on denoising score matching (DSM) Ho et al. (2020); Song et al. (2021b), which estimates the gradient of the log-density of perturbed data distributions. The original formulation establishes a robust probabilistic foundation and facilitates maximum likelihood training through a variational lower bound. Recent studies have advanced the theoretical understanding of score matching within diffusion models. Han *et al*. Han et al. (2024) established generalization error bounds for neural network-based score estimation, providing insights into the optimization and generalization aspects of learning the score function. Dou *et al*. Dou et al. (2024) derived sharp minimax rates for score estimation, emphasizing the statistical efficiency of score matching under smoothness assumptions. Furthermore, Liang *et al*. Liang et al. (2025) analyzed the convergence behavior of diffusion models under score mismatches, providing theoretical guarantees for zero-shot

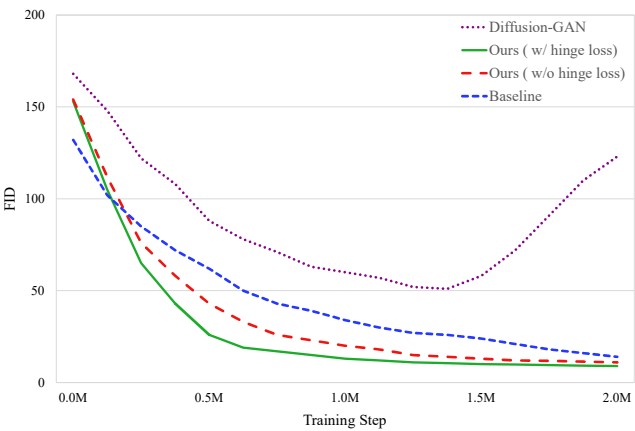

Figure 7: Training convergence and stability comparison for ablation study.

conditional sampling scenarios. Shen and Oates Shen & Oates (2024) introduced operator-informed score matching, utilizing Markov diffusion processes to improve training efficiency.

While these perspectives enhance our understanding of the learning objective, the previous methods do not address how adversarial regularization interacts with score-based training at a theoretical level. Our work bridges this gap by formulating DSL training as functional gradient descent, regularized by a margin-based energy landscape. We show that the equilibrium of the DSL game is consistent with the true score function under mild assumptions, thereby ensuring reliability even in the presence of an active adversarial margin. This theoretical consistency distinguishes DSL from prior adversarial approaches, which often introduce auxiliary objectives without ensuring alignment with the target score. As a result, DSL provides both theoretical robustness and practical improvements in fidelity while maintaining the integrity of the original training paradigm of diffusion models.

## B    ABLATION STUDY

In this section, we performed a comprehensive analysis of the convergence speed and stability of the proposed method by comparing it with baseline methods. We adopted Stable Diffusion Podell et al. (2024); Rombach et al. (2022) as the backbone diffusion model and used Diffusion-GAN Wang et al. (2023) as the baseline for adversarially trained diffusion models. Diffusion-GAN introduces an additional discriminator into the latent denoising process and employs alternating training with the generator (*i.e.*, denoising diffusion network). In contrast, the proposed method integrates an energy-based discriminator with a margin-based hinge loss to adaptively reweight gradients, thereby enhancing score estimation. This allows for the seamless integration of adversarial training to be seamlessly incorporated into the diffusion pipeline without requiring alternating optimization or redesigning the sampling process. To isolate the effect of the hinge loss, we conducted ablation studies that compare the scenarios with and without margin-based regularization. As described in Sec. 3.1, we adopted a standard LoRA Hu et al. (2022) for the discriminator. The discriminator introduces one additional forward pass through the frozen backbone during training. Thus, the discriminator shares a single SDXL UNet backbone ($\sim$2.6B) of the generator and requires only a small number of additional LoRA parameters ($\sim$24M) on the discriminator side. This design keeps the number of trainable parameters and activations for backpropagation relatively small for the discriminator ($\leq$ 1%). Under the same GPU setting, this increases the wall-clock time per training step by 19.3%.

For a fair comparison, all methods employed the same denoising backbone (*i.e.*, SDXL Podell et al. (2024)) and are trained from scratch on an identical subset of the LAION-5B Schuhmann et al. (2022) dataset using the same training steps and parameter budgets. Following prior work Zhang et al. (2023); Rombach et al. (2022), we used the Fréchet Inception Distance (FID) Heusel et al. (2017) measured at regular intervals to assess convergence speed and training stability.

Figure 7 presents the evolution of FID scores over 2M training steps. While Diffusion-GAN initially shows a declining trend in FID, it begins to diverge after approximately 1.4M steps, reflecting

Table 5: Comparisons with respect to $\lambda$ values.

| $\lambda$ | FID ($\downarrow$) | CLIP score ($\uparrow$) |
|---|---|---|
| 0.1 | 10.89 | 0.3218 |
| 0.05 | **9.21** | **0.3481** |
| 0.01 | 12.32 | 0.3194 |

instability in adversarial training. This is attributed to the limitations of binary classification-based discriminators in capturing the complex and multi-modal distribution of large-scale datasets like LAION-5B. As a result, the generator and discriminator updates become unbalanced, leading to instability and potential mode collapse.

In contrast, the proposed method with hinge loss consistently decreases FID and achieves FID of 9.32 after 1.2M training steps and converges stably. This is enabled by the energy-based discriminator and margin-based regularization, which reweight gradients adaptively based on sample difficulty. The discriminator thus provides informative feedback beyond binary classification, guiding the generator to focus more effectively on challenging regions of the data distribution. Consequently, our method mitigates training instability and avoids mode collapse, even on diverse and complex datasets.

However, the variant without hinge loss, while still convergent, exhibits mild oscillations in the FID trajectory. This suggests that without margin-based reweighting, gradient updates lack adaptivity, leading to unstable learning on harder samples. The model captures the coarse distribution but struggles with fine-grained detail, resulting in fluctuating convergence behavior and reduced learning efficiency.

In contrast, the non-adversarial baseline, i.e., Stable Diffusion, attains an FID of 13.81 only after 2.0M steps. This demonstrates that our method achieves significantly faster convergence and lower FID scores compared to the baseline. Specifically, DSL requires ~40% fewer training steps to reach convergence, which yields a ~28.4% reduction in total training time while simultaneously improving the FID by ~32.5%. These improvements validate that the energy-based adversarial framework enhances both the convergence rate and overall training efficiency. Specifically, the margin-based hinge loss facilitates self-correcting regularization, directing the model to focus more on difficult samples. By adaptively reweighting gradients early in training, the proposed method accelerates convergence and improves generalization to complex distributions, offering stable and efficient learning behavior that surpasses existing baselines.

## C   SENSITIVITY ANALYSIS OF $\lambda$

In this section, we analyzed the sensitivity or the proposed method with respect to the adversarial weight $\lambda$ defined in (2). To ensure a fair comparison, all experiments were trained on the same subset of the LAION-5B Schuhmann et al. (2022) dataset. Except for the value of $\lambda$, all other hyperparameters and training configurations were kept constant as specified in Section 3.1. Following the same evaluation protocol as in Sec. 3.2.1, we randomly selected 20K image-text pairs from the COCO2014 validation set Lin et al. (2014) and the Flickr30k dataset Young et al. (2014). Both FID and CLIP scores were used to assess the quality of generated images.

We conducted comparative experiments with three different values of $\lambda$ : $0.1, 0.005, 0.001$. The results are summarized in Table 5. The empirical results demonstrate that $\lambda = 0.05$ produces the best performance in terms of both FID and CLIP score, indicating an optimal trade-off between adversarial and denoising objectives.

When $\lambda$ is set too high (*e.g.*, 0.1), the discriminator's hinge-style margin loss imposes a strict separation between real and generated noise. This causes the adversarial signal to dominate the training, causing the discriminator to overpower the generator. As a result, the generator becomes overly reliant on adversarial feedback, which impairs its ability to retain denoising performance and effectively model the true data distribution. This imbalance degrades the quality and diversity of the generated samples and often introduces subtle artifacts.

Conversely, when $\lambda$ is too low (*e.g.*, 0.01), the adversarial signal becomes too weak, diminishing the discriminator's impact. In this regime, the generator focuses primarily on minimizing reconstruction loss, making the training process similar to that of a conventional denoising autoencoder without

adversarial guidance. Consequently, the perceptual sharpness of the generated samples deteriorates and improvements over the baseline become marginal. These findings underscore the importance of appropriately tuning the adversarial weight $\lambda$ to maintain both the fidelity and diversity of the generated samples. Striking the right balance ensures that the generator benefits from meaningful adversarial feedback without compromising its denoising capability.

Moreover, we observed similar trends across various generative tasks, including conditional text-to-image generation, image super-resolution, and 2D-to-3D reconstruction. In all these settings, using $\lambda = 0.05$ consistently led to strong performance without the need for task-specific retuning. We attribute this robustness to the self-correcting nature of our proposed energy-based regularization, which allows the discriminator to focus on more challenging regions of the data distribution. This mechanism prevents the adversarial signal from overwhelming the training or becoming ineffective, thus promoting stable learning dynamics across diverse domains. These empirical observations suggest that a single $\lambda$ value generalizes well across multiple generation tasks, highlighting the adaptability and robustness of our approach. In practical terms, this eliminates the need for extensive hyperparameter search, making the method both effective and efficient in real-world applications.

## D  ADDITIONAL QUALITATIVE COMPARISONS

Due to page constraints of the main manuscript, we provide additional qualitative comparisons to further validate the effectiveness of our proposed method across diverse generation tasks, including text-to-image generation, conditional text-to-image generation, image super-resolution, and 2D-to-3D reconstruction. The results extend the main qualitative comparisons discussed in Sec. 3.2 of the main paper, demonstrating the effectiveness of the proposed method in improving perceptual fidelity, structural consistency, and sample diversity.

### D.1  TEXT-TO-IMAGE GENERATION

Figure 8 presents additional comparison results for the text-to-image generation task, comparing the proposed method with Stable Diffusion Rombach et al. (2022). These results supplement the findings discussed in Sec. 3.2.1. These examples further demonstrate that the proposed method consistently generates images with sharper detail and more accurate semantic alignment to the input prompts, particularly in the context of challenging compositions or uncommon visual descriptions.

In Fig. 8 (a), for the prompt "a glowing cat sitting on top of a floating streetlight in the middle of the ocean", the proposed method produces a vivid and realistic representation, while the baseline fails to convincingly depict the structural elements of the scene. Similarly, in Fig. 8 (d), while both methods capture the "a pink lake at sunset" background reasonably well, the baseline omits the "a giant teacup", which is semantically central to the prompt. In contrast, our method preserves the full intent of the input text, generating a well-balanced and coherent image. For instance, in Fig. 8 (l), with a rare concept such as "a library inside a giant seashell", the baseline omits key elements, while our method generates a plausible and semantically rich scene with spatial coherence. Similarly, in Fig. 8 (n), the proposed method accurately renders both "a castle" and "a sleeping cat" as distinct, well-structured elements, while the baseline fails to generate the cat altogether.

Our method also demonstrates stronger performance in rendering complex human figures and expressions. In Fig. 8 (e), the baseline produces an ambiguous body structure for "a girl playing an acoustic guitar", with artifacts and blurred features. In contrast, our method renders anatomically correct body parts, facial expressions, and even harmonizes the background (sunset) with the main subject. Likewise, Fig. 8 (i) shows that the proposed method captures every component of the prompt "A girl playing with a dog in a lush green garden with flowers blooming", from the interaction between the characters to the vivid setting, with perceptual realism and accurate object placement, unlike the baseline, which generates distorted anatomy and inconsistent layout. These significant differences also extend to artistic styles. In Fig. 8 (c), the baseline struggles to depict the structure of "a fox wearing a scarf, riding a bicycle", failing to render the motion and body alignment. Our method captures both the posture and accessories of the character with clarity, resulting in a more expressive and believable illustration. Similarly, Fig. 8 (f) demonstrates that our method better preserves the structure and high-frequency details of "a rainy city street reflected in puddles", including reflections, buildings, and vehicle contours.

In general, these results underscore the effectiveness of our approach in enhancing sample fidelity, perceptual realism, and semantic consistency. By introducing Discrepancy-aware Score Learning

(DSL) into the diffusion framework, the proposed method achieves superior structural alignment and image quality across a wide range of content types and visual styles.

## D.2    CONDITIONAL TEXT-TO-IMAGE GENERATION

Figure 9 presents supplementary qualitative comparisons between our proposed method and GLI-GEN Li et al. (2023), complementing the results presented in Sec. 3.2.2. These results underscore the efficacy of the proposed method in generating semantically accurate and visually coherent images that are precisely aligned with both the input text and the specified bounding boxes.

In Fig. 9 (a), the baseline method effectively identifies the approximate locations of the bounding boxes; however, it does not adequately represent the textual content. For instance, the skateboard is not clearly generated, and the scarf on the dog appears green rather than red, suggesting weak semantic grounding. In contrast, our method successfully renders both "a skateboard" and "a red scarf" with accurate spatial and visual fidelity, demonstrating superior alignment between box and text modalities. Similarly, in Fig. 9 (c), both methods attempt to generate "a robot holding balloons on a sidewalk next to a streetlamp". However, the baseline version lacks definition and realism in the robot figure, with vague forms and poor object distinction. Our method, on the other hand, produces a well-defined and high-quality robot, capturing small but critical details such as limbs and facial elements, which collectively enhance perceptual realism and reinforce semantic consistency.

The performance gap becomes even more pronounced in more complex scenes involving overlapping boxes or highly imaginative prompts. For instance, in Fig. 9 (d), where the prompt involves "a street artist with paint cans on a small table, painting on a canvas", the baseline model fails to render key elements such as the paint cans and produces distorted human features. In contrast, our method faithfully generates all objects mentioned in the text, maintaining proper box alignment and preserving fine-grained anatomical details such as arms and hand posture. The spatial arrangement of all components appears harmonious and coherent, contributing to the overall image realism. Further, in Fig. 9 (e), which involves the imaginative concept of "a frog wearing a tiny crown sitting on a lily pad, holding an umbrella under the rain", the baseline output lacks structure and realism. The proposed method excels here by producing a photorealistic and structurally sound image, accurately depicting each specified object with appropriate shape and spatial relationships. Notably, no aspect of the text or bounding box input is omitted, and the composition is perceptually natural.

Overall, these results validate the effectiveness of our approach in not only aligning bounding boxes with text but also achieving superior semantic fidelity and structural realism. By incorporating our adversarial training framework based on discrepancy-aware learning, the proposed method enables better integration of conditional constraints and improves object quality, spatial consistency, and perceptual detail in complex and constrained generation scenarios.

Moreover, Fig. 10 illustrated additional qualitative comparisons between the proposed method and Textual Inversion Gal et al. (2022a), using various prompts to evaluate the ability of each method to generalize learned concepts across styles and compositions. The results highlight the proposed method's superior adaptability, visual fidelity, and semantic consistency under low-data conditions.

Specifically, in Fig. 10 (a), the proposed method successfully renders "a pixel art of $S_*$" while preserving the identity of the input concept and adapting it to the stylistic transformation. In contrast, the baseline method overfits to the original sample and fails to reflect the "pixel art" style, resulting in output that lacks visual abstraction and fusion with the prompt. A similar trend is observed in Fig. 10 (b), where the proposed method naturally integrates "blue overalls" into the concept while maintaining object identity. The baseline, however, fails to adapt due to overfitting, leading to a rigid replication of the input and semantic inconsistency with the textual prompt.

This overfitting issue becomes more apparent as the visual complexity of the input increases. For example, in Figs. 10 (d), (e), and (f), the baseline repeatedly generates nearly identical outputs regardless of whether the prompt involves "a watercolor painting", "a sofa", or "a green $S_*$". The lack of variation reveals its limited capacity to disentangle style from concept. By contrast, the proposed method retains the visual identity of $S_*$ while naturally blending it with diverse stylistic or semantic cues, producing unique and coherent images aligned with each prompt. Further examples in Figs. 10 (g) and (h) demonstrate how our method handles more complex prompts like "a vase looks like $S_*$ with flowers" or "a movie poster of $S_*$". The proposed method effectively incorporates additional visual elements (*e.g.*, flowers and poster layout) while preserving the concept's structure and visual coherence. In contrast, the baseline fails to render key components, such as the flowers in

(g), or to adjust the composition to match a poster format in (h), highlighting its limited generative flexibility.

These results collectively demonstrate the strong generalization ability of the proposed method, even in low-data regimes. By incorporating adversarial training with margin-based energy minimization into the diffusion process, our method avoids overfitting and achieves stable convergence. It also enables the generation of images that are both diverse and perceptually coherent across varying captions and styles. This robustness in semantic blending and visual adaptation underscores the practicality of our approach for real-world applications involving concept-driven generation.

### D.3 IMAGE SUPER-RESOLUTION

Figure 11 presents additional qualitative comparisons for the image super-resolution task, complementing the findings presented in Sec. 3.2.3 We compared the proposed method with StableSR Wang et al. (2024a). The results consistently demonstrate that our method excels at recovering spatial structures and surface details with high fidelity, producing sharper and more realistic images overall.

In the first example of Fig. 11, our method more clearly reconstructs ceiling textures and road edges, which appear blurry in the output of the baseline method. Similarly, in the second sample, while the baseline struggles with clearly rendering vehicle contours, lane markings, and the separation between buildings, the proposed method produces a more distinct and spatially coherent output. The improved reconstruction includes well-defined road lines and sharper building silhouettes. The performance gap widens as the complexity of the scene increases. For instance, in the third sample, which features multiple overlapping elements such as vehicles, roads, buildings, and trees, the baseline partially restores object boundaries but fails to recover fine textures or surface variations. In contrast, our method not only preserves structural consistency but also reconstructs detailed patterns on different objects while maintaining their distinct visual identities. This advantage extends to the fourth and fifth samples. The proposed method captures intricate surface patterns and textures more accurately than the baseline, while also reconstructing lane markers and road boundaries with high precision. Such fine-grained details are essential for maintaining perceptual sharpness and realism in high-frequency regions of the image.

These results highlight the effectiveness of the proposed method in capturing both global spatial structures and localized surface details, leading to perceptually superior high-resolution outputs. By incorporating Discrepancy-aware Score Learning (DSL) into the diffusion framework, our method achieves robust and stable reconstruction, even for complex and low-resolution inputs. It consistently generates images that are visually sharp, structurally coherent, and high in fidelity, outperforming baseline diffusion-based super-resolution models in both clarity and realism.

### D.4 2D-TO-3D GENERATION

Figure 12 presents additional qualitative comparisons between our proposed method and Sync-Dreamer Liu et al. (2023b) for the 2D-to-3D generation task. These examples expand upon the evaluations discussed in Sec. 3.2.4, demonstrating that the proposed method achieves superior multi-view consistency and reconstructs textured meshes with more accurate geometry and richer detail across diverse scenarios.

In the first example, the baseline method struggles to capture structural elements such as stairs and windows in the multi-view images, leading to a blurred and overly simplified representation. These weaknesses are further reflected in the reconstructed mesh, which lacks geometric precision and appears over-smoothed. In contrast, the proposed method effectively preserves fine-grained architectural features, including chimneys, window frames, and floral elements, across multiple views. Additionally, it reconstructs a well-defined 3D mesh characterized by sharper contours and enhanced structural fidelity. The second example showcases an animal figure with detailed surface features. The baseline method fails to preserve local texture, rendering the fur and facial features indistinct, while the mesh lacks definition and symmetry. Our method, however, maintains the integrity of key details such as the eyes, ears, and tail in the multi-view images, producing a mesh that is both geometrically accurate and perceptually realistic.

These advantages become more pronounced in the third example, where the input image provides limited depth cues. Here, the baseline's reconstruction flattens out the object in side views, indicating a failure to infer volumetric form. Our method successfully recovers the underlying 3D structure,

maintaining volume and spatial coherence even from oblique angles. Finally, in the fourth example, the baseline produces a flat and indistinct reconstruction. In contrast, our method preserves both geometric complexity and textural richness. This results in a multi-view output characterized by high visual consistency and a clean, well-articulated mesh.

Overall, these results confirm the robustness and effectiveness of the proposed method in 2D-to-3D generation. By incorporating our adversarial training framework into the diffusion process, our model generates high-fidelity multi-view images that are consistent across viewpoints—even from ambiguous or flat inputs—and enables high-quality 3D reconstruction. This is accomplished through improved learning of geometric structures and detailed surface textures, ultimately yielding perceptually compelling and structurally accurate 3D outputs that exceed those of existing methods.

Moreover, Fig. 13 presents additional qualitative comparisons between the proposed method and Era3D Li et al. (2024a), providing further evidence of the generalizability and robustness of our approach for 2D-to-3D generation. The results indicate that our method consistently preserves both geometric and perceptual consistency across multiple views while effectively capturing structural integrity and fine-grained details. This capability significantly enhances the quality of 3D reconstruction.

Specifically, in the first sample of Fig. 13, the baseline method exhibits difficulty in maintaining structural coherence across the multi-view images. The legs of the object appear to be misaligned, and the facial textures, along with local details, such as surface features, are blurred or distorted. This inconsistency leads to a textured mesh that is overly smoothed and lacks distinct facial structure or fine surface characteristics. In contrast, our method preserves both the geometric form and detailed appearance across views. A comparable advantage is evident in the second example. Our method produces well-defined multi-view images with high-frequency details such as wheel textures and the geometric structure of the driver figure. The corresponding reconstruction demonstrates precise block structures in the ceiling and wheels, indicating a high level of consistency in both geometry and appearance. In comparison, the baseline method fails to reconstruct these details clearly, resulting in a less expressive and flatter mesh.

These differences are even more pronounced with complex input images. In the third example, the multi-view images produced by the baseline are blurry and lack clarity. Crucial local features such as the flower pattern on the torso are omitted, and the number of horns is incorrectly duplicated, indicating inconsistent view synthesis. These errors propagate into the mesh, where misaligned hair and torso regions show visible artifacts. Our method, in contrast, accurately captures all semantic elements, including the flower and horn structure, across consistent and sharp views. This leads to a complete and perceptually faithful 3D reconstruction. In the final example, the baseline method produces implausible geometries, such as a tree branch intersecting the bird's torso. Moreover, it introduces texture artifacts in both the bird's feathers and the surrounding flowers. The proposed method addresses these issues by generating multi-view images that exhibit realistic geometry and fine-grained texture, thereby accurately representing the branch, bird posture, and flower details. Taken together, these results demonstrate the efficacy of the proposed method in capturing both global geometry and local texture across multiple views. This approach yields multi-view images that are semantically aligned and perceptually coherent. By integrating adversarial training with margin-based energy minimization into the diffusion framework, our method achieves high-fidelity 3D reconstruction from complex or ambiguous input images. This approach demonstrates superior performance compared to previous methods in terms of structural accuracy and detail preservation.

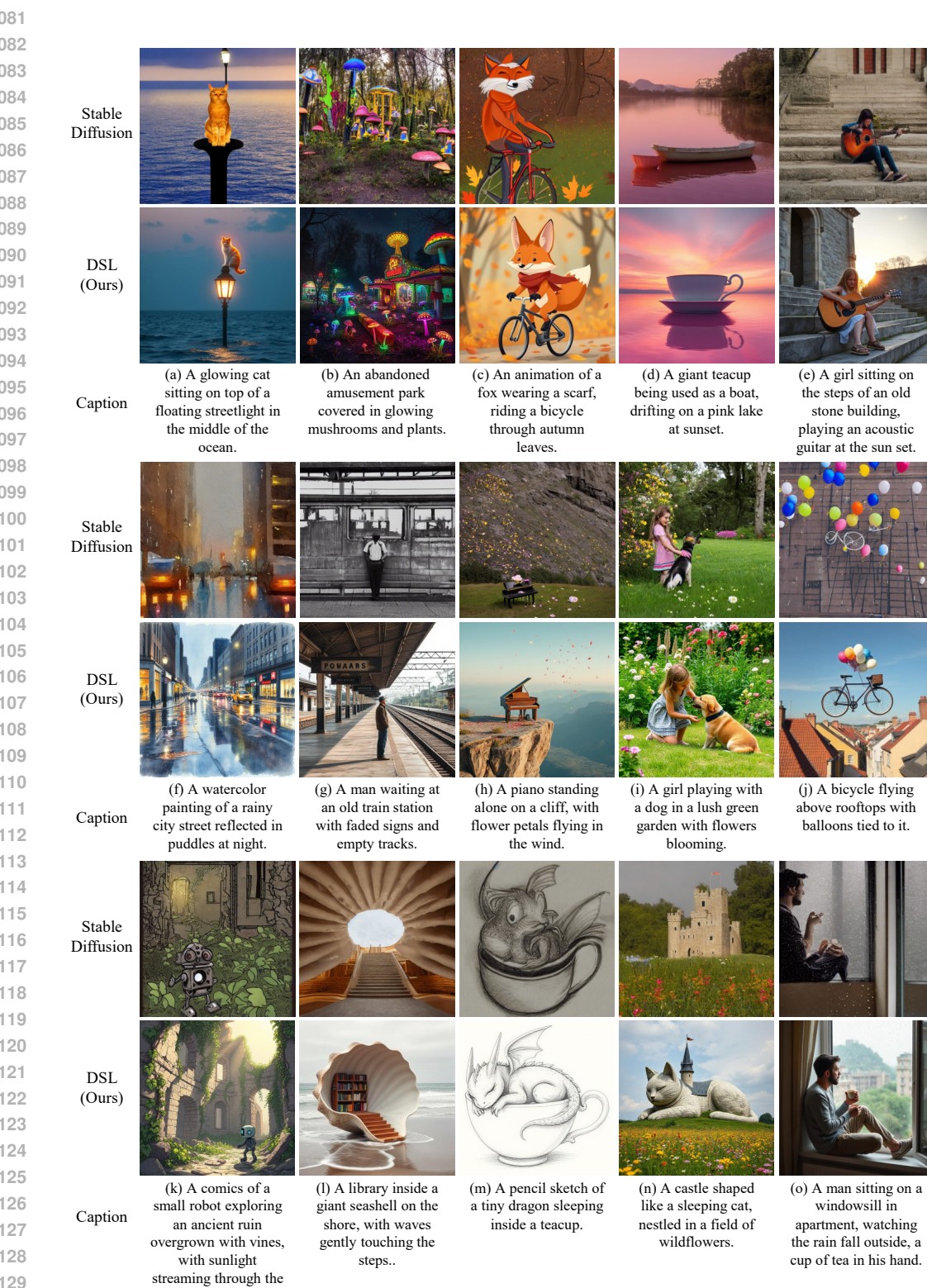

Figure 8: Additional qualitative comparisons with Stable Diffusion.

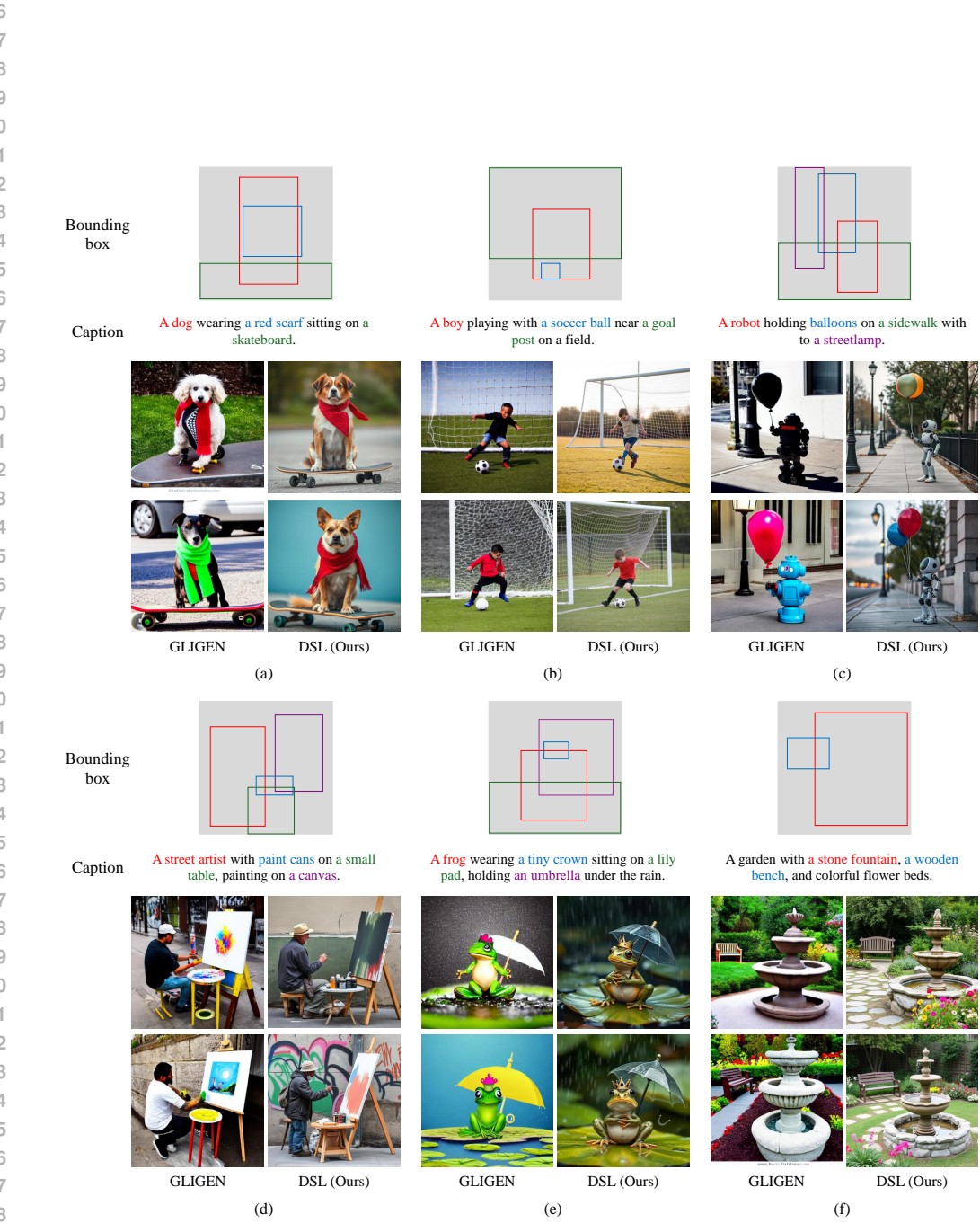

Figure 9: Additional qualitative comparisons with GLIGEN.

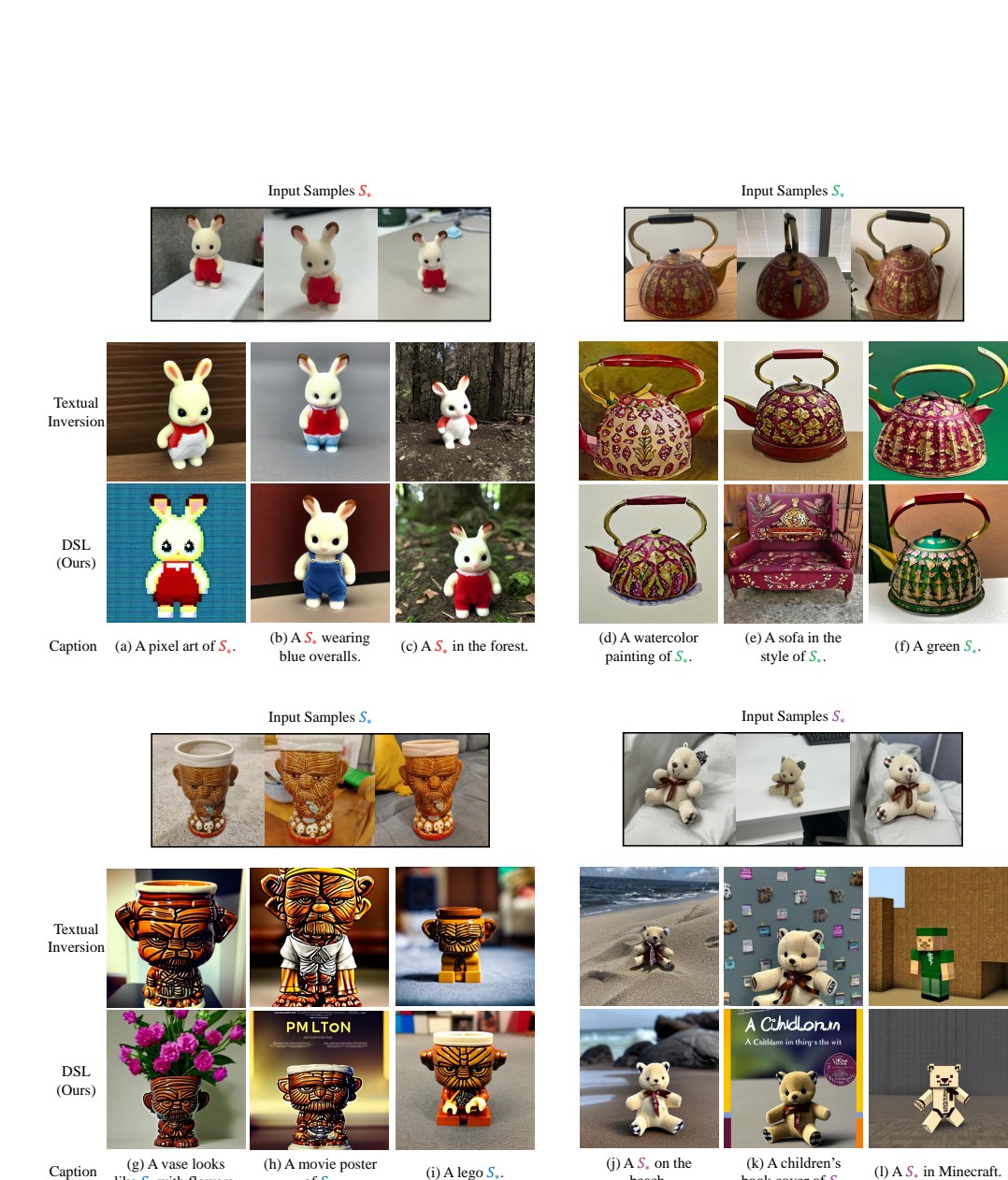

Figure 10: Additional qualitative comparisons with Textual Inversion.

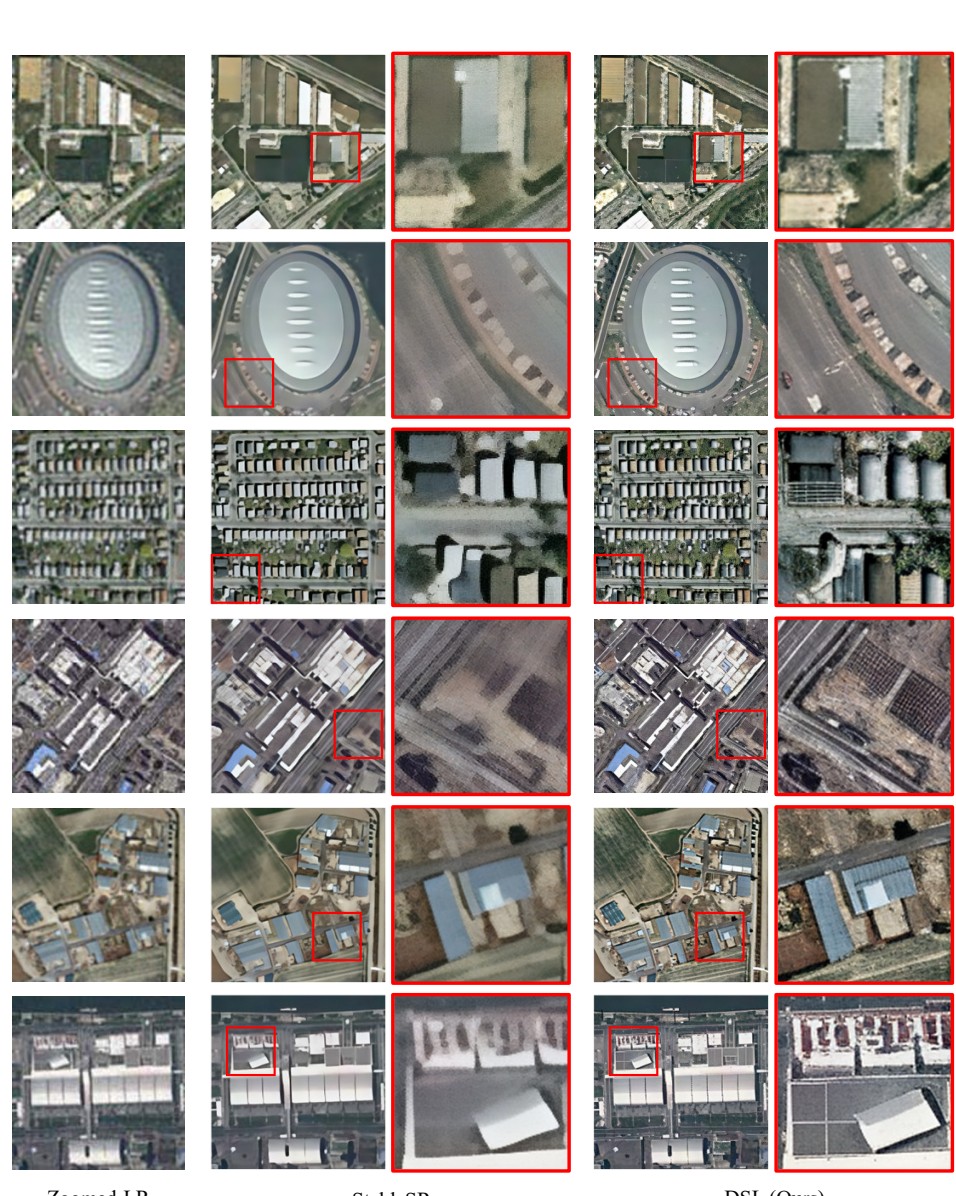

Zoomed LR        StableSR        DSL (Ours)

Figure 11: Additional qualitative comparisons with StableSR.

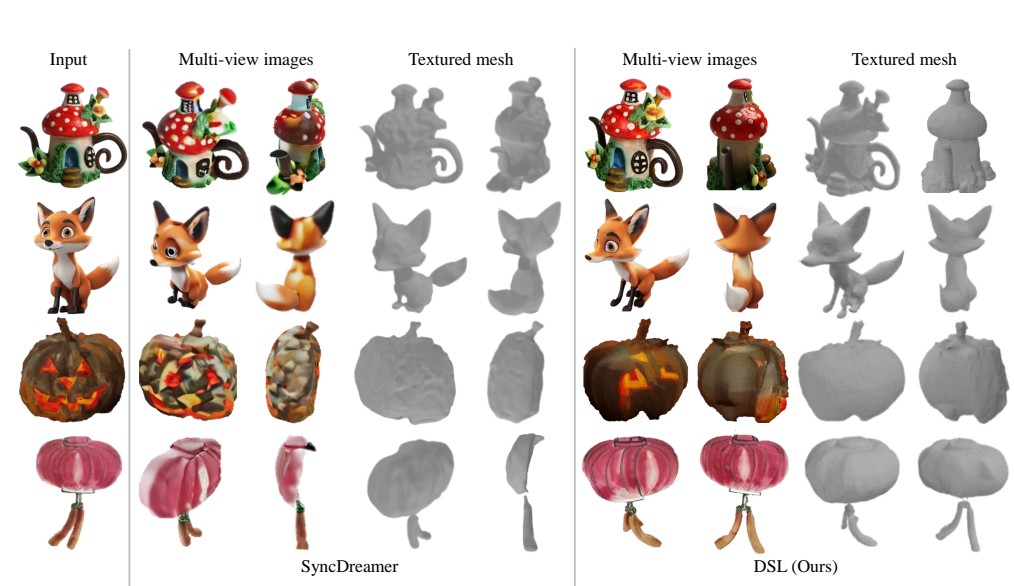

Figure 12: Additional qualitative comparisons with SyncDreamer.

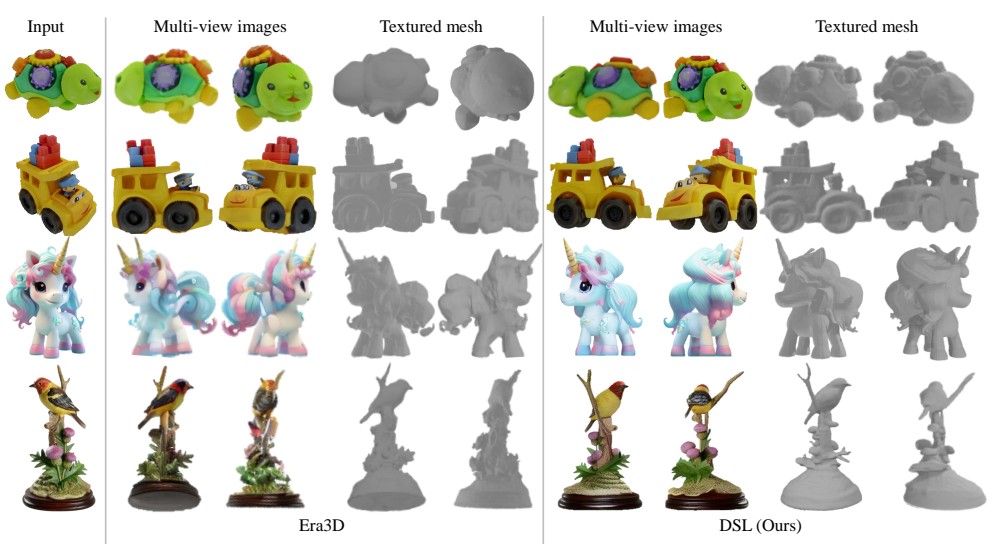

Figure 13: Additional qualitative comparisons with Era3D.