# OpenReview forum: "Discrepancy-aware Score Learningfor Diffusion Training"
_ICLR.cc/2026/Conference — Submitted to ICLR 2026_

### Official Review · Reviewer_Mgsj · 2025-10-26

**Soundness:** 2
**Presentation:** 2
**Contribution:** 2
**Rating:** 2
**Confidence:** 3

**Summary:**

This paper proposes Diffusion Score Learning (DSL), a new training framework for diffusion models. The method augments the standard Denoising Score Matching (DSM) objective by introducing a discriminator ($D_\phi$). The core idea is to provide a more "robust" supervisory signal using a hinge-based loss function:$L_\phi = |\epsilon_\phi - \epsilon|^2 + \lambda \max(0, m - |\epsilon_\phi - \epsilon_G|^2)$The authors claim this design emphasizes hard samples, reduces the variance of the supervision target, and enhances overall training stability.

**Strengths:**

Clear Motivation: The paper is well-motivated, addressing the critical and widely-recognized issues of high target variance and training instability in standard diffusion model training.

Framework Simplicity: The proposed framework is simple and presents a natural, clean integration of GAN-based adversarial learning with the score-matching objective.

**Weaknesses:**

Fundamental Contradiction in "Hard-Sample Emphasis" Claim: The paper's central claim of 'hard-sample emphasis' appears to be in direct contradiction with its own mathematical formulation. The hinge term, $\lambda \max(0, m - |\epsilon_\phi - \epsilon_G|^2)$, only provides a non-zero gradient when the squared error $|\epsilon_\phi - \epsilon_G|^2$ is less than the margin $m$. For "hard samples," which are presumably those with a large error (i.e., $> m$), the gradient from this term is exactly zero. This mechanism seems to explicitly ignore hard samples and focus only on 'easy' samples (those already within the margin), which is the opposite of the stated motivation.

Unverified Variance Reduction Hypothesis: The paper posits that the discriminator provides a more stable, lower-variance supervisory signal (e.g., $\epsilon_G$) than the original noise target $\epsilon$. This is a core hypothesis, yet it is not empirically validated. The paper lacks a essential quantitative analysis comparing the variance of the discriminator's output (Var($\epsilon_G$)) with the variance of the ground-truth noise (Var($\epsilon$)).

Lack of Ablation and Stability Analysis: The paper fails to investigate the impact of its key hyperparameters, namely the margin $m$ and the weighting $\lambda$, on the model's convergence and stability. It is unclear how sensitive the training is to these parameters, how they should be set, or what failure modes (e.g., oscillations, divergence) might arise.

Limited Novelty: The proposed loss function bears a strong resemblance to the margin-based loss used in EBGAN. The paper does not clearly articulate the novel theoretical contributions or insights that differentiate DSL from this well-established prior work.

Questionable Scalability: The reliance on a discriminator-based architecture, which is notoriously difficult to stabilize (e.g., mode collapse, training dynamics), raises significant concerns about the method's scalability. Its viability for larger datasets and higher-resolution image generation tasks remains unproven and questionable.

**Questions:**

Given that the hinge term provides zero gradient for samples with an error $|\epsilon_\phi - \epsilon_G|^2 > m$, could the authors please clarify their claim of "hard-sample emphasis"? It appears to mechanically do the opposite.

Does the hinge term introduce a systematic bias by pushing the optimization target away from the true noise $\epsilon$? Is there a risk of the model's output distribution drifting if this margin-based objective is not carefully balanced?

Can the authors provide a quantitative analysis (e.g., plotting the variance over time) to support the claim that the discriminator's output $\epsilon_G$ actually has a lower variance than the original noise target $\epsilon$?

What is the impact of the margin $m$ and $\lambda$ on training stability and final performance? How were these values chosen, and did the authors observe any convergence issues?

Could the authors elaborate on the substantial theoretical or empirical differences between DSL and the margin-based loss in EBGAN, beyond the application to diffusion models?

---

> ### Author Response · Authors · 2025-11-21
> **Response to Reviewer Mgsj (1/3)**
>
> Thank you for your thoughtful comments. Your feedback helped us better articulate motivation and clarify key formulations in our method.
> ‎
> >**Q1. Given that the hinge term provides zero gradient for samples with an error $|\epsilon_\phi - \epsilon_G|^2 > m$, could the authors please clarify their claim of "hard-sample emphasis"? It appears to mechanically do the opposite.**
>
> >**Q2. Does the hinge term introduce a systematic bias by pushing the optimization target away from the true noise $\epsilon$? Is there a risk of the model's output distribution drifting if this margin-based objective is not carefully balanced?**
>
> **A1,2.** Thank you for the insightful questions about the hinge loss and its effect on hard-sample emphasis and potential bias.
> We agree that if the margin $m$ is set too small, the hinge term might not give a gradient for large error samples and, thus, cannot emphasize harder examples. On the other hand, if $m$ is set too large, the discriminator becomes too sensitive to noise and outliers, which can destabilize training and can push the model away from the true noise distribution.
>
> Our design goal is to emphasize challenging but learnable samples, rather than all large-error samples including pathological outliers. To achieve this, in our implementation we adaptively set the margin $m$ to the empirical mean error based on a decision-theoretic criterion, as described in Sec. 3.1. In our implementation, we periodically ran inference on the training set (*e.g.*, every five epochs), computed the mean squared discrepancy, and set $m$ to this mean error. In detail,
> * Samples with very small errors are already well explained by the generator and receive relatively weaker influence from the hinge term.
> * Samples with moderately large error around the mean error are actively penalized by the hinge, so the discriminator focuses on regions where the generator still significantly differs from the teacher but is not dominated by extreme outliers.
> * Samples with extremely large errors, which are often due to noise, rare or corrupted data, or highly unlikely configurations, are downweighted by the zero gradient of the hinge term and therefore contribute less to instability.
>
> Therefore, DSL allocates more adversarial pressure to samples near the data-dependent decision boundary between generator output and real noise, rather than to all samples with arbitrarily large errors. In other words, the hinge loss, together with the adaptive margin, emphasizes hard informative examples while suppressing uninformative outliers.
>
> Regarding potential bias and distributional drift, the discriminator loss is defined in Eq. (2) of the manuscript, and the generator minimizes Eq. (3).
> At the ideal equilibrium where $\epsilon_\phi = \epsilon_G = \epsilon$, both the MSE term and the hinge term vanish. This equilibrium coincides with the standard DSM optimum. Our theoretical analysis in Sec. 2 shows that DSL can be interpreted as a regularized score matching objective and that the DSM fixed point remains a stationary point of the Wasserstein gradient flow induced by DSL. Thus, the margin-based term does not introduce a systematic shift of the target away from the true noise in the idealized limit.
>
> In practice, the adaptive choice of $m$ is crucial to avoid undesirable bias. We performed a sensitivity experiment on the margin and observed that varying $m$ within $\pm$ 10$% of the mean error does not yield significant performance differences, while deviations larger than $\pm$ 20% lead to unstable training and degraded performance. These results suggest that the empirical mean error provides a reasonable operating point that balances stability and hard-sample emphasis.
>
> While our experiments show that the mean-error based margin works well and does not cause noticeable drift in the output distribution in the evaluated settings, we agree that other margin formulations or learnable margin schemes may further improve the trade-off between bias and stability. We will release our full training code so that different margin designs and optimization strategies can be systematically explored and validated.
>
> ---

---

> > ### Author Response · Authors · 2025-11-21
> > **Response to Reviewer Mgsj (2/3)**
> >
> > ---
> > >**Q3. Can the authors provide a quantitative analysis (e.g., plotting the variance over time) to support the claim that the discriminator's output $\epsilon_G$ actually has a lower variance than the original noise target $\epsilon$?**
> >
> > **A3.** We appreciate the constructive suggestion regarding variance analysis.
> > We agree that a temporal variance analysis is important to support our claim that the discriminator output $\epsilon_G$ provides a lower-variance supervisory signal than the original noise target $\epsilon$. We have started additional training runs to collect such statistics over the course of training. However, due to the computational cost of these large-scale experiments and the limited rebuttal period, it is not feasible to complete a full and reliable variance study for all settings within the review timeline.
> >
> > Therefore, to reflect the reviewer’s comments and clarify our claim, we will include and discuss the empirical variance trajectories of $\epsilon$ and $\epsilon_G$ over training time in the supplemental material at the camera-ready submission.
> >
> > ---
> >
> > >**Q4. What is the impact of the margin $m$ and $\lambda$ on training stability and final performance? How were these values chosen, and did the authors observe any convergence issues?**
> >
> > **A4.** Thank you for raising this important question about hyperparameter sensitivity.
> > For the margin $m$, please refer to our response in **A1,2**. In summary, $m$ is determined adaptively using a mean-error criterion grounded in decision-theoretic principles, which yields a closed-form update without extra training, and our sensitivity analysis shows that moderate perturbations around this value do not significantly degrade performance, while very large deviations can lead to instability.
> >
> > For the weighting factor $\lambda$, we performed a grid search over several candidates in our main experiments. As detailed in Sec. C of the supplemental material, we found that $\lambda = 0.05$ consistently yields the best or near-best performance across various tasks, such as text-to-image, 2D-to-3D, and super-resolution generations. Importantly, the performance curves as a function of $\lambda$ vary smoothly without signs of abrupt divergence or mode collapse, which suggests that DSL is reasonably robust to moderate changes in $\lambda$.
> >
> > Taken together, the empirical evidence indicates that there exists a stable regime where a single choice of $\lambda$ and the mean-error-based margin $m$ works well across different datasets and tasks. We will release our code and configuration files so that others can reproduce our sensitivity studies and explore different configurations of $m$ and $\lambda$ under their own experimental setups.
> >
> > ---

---

> ### Author Response · Authors · 2025-11-21
> **Response to Reviewer Mgsj (3/3)**
>
> Response to Reviewer Mgsj (3/3)
>
> ---
>
> >**Q5. Could the authors elaborate on the substantial theoretical or empirical differences between DSL and the margin-based loss in EBGAN, beyond the application to diffusion models?**
>
> **A5.** Thank you for this detailed question. We summarize the main differences.
>
> First, the core contribution of our work is:
> 1. to propose Discrepancy-aware Score Learning (DSL), which extends standard Denoising Score Matching (DSM) for diffusion models by introducing an energy-based discriminator in the noise space, and
> 2. to reformulate DSL as a margin-based energy minimization problem that generalizes DSM into a discrepancy-aware reweighting scheme, with an accompanying analysis as functional gradient descent in the space of probability distributions.
>
> In contrast, DSL defines the energy entirely in the noise space. The discriminator compares the predicted noise $\epsilon_\phi$ with both the ground truth noise $\epsilon$ and the generator prediction $\epsilon_G$, and the margin-based hinge term is applied to $|\epsilon_\phi - \epsilon_G|^2$. This design makes the adversarial signal act directly on the noise prediction error, which is the primary training target of diffusion models, rather than on a separate reconstruction metric in pixel space.
>
> Second, beyond this change of energy domain, we analyze DSL from the perspective of Wasserstein gradient flows and DSM equilibrium. We show that under mild assumptions, the joint minimizer $(\epsilon_G^\star, \epsilon_\phi^\star)$ satisfies $\epsilon_\phi^\star = \epsilon$ and $\epsilon_G^\star$ lies in a bounded neighbourhood around the true noise, and that DSL reduces to DSM in the appropriate limits (e.g., $m \to 0$ or $\epsilon_\phi \to \epsilon$). This provides a theoretical connection between the adversarial, margin-based training and the original score matching objective of diffusion models, which is not present in the original EBGAN analysis.
>
> In this sense, DSL does not simply reuse the EBGAN margin-based loss. It relocates the energy definition from image space to noise space, tightly couples the adversarial term with the score matching objective, and provides a functional gradient and equilibrium analysis specific to denoising diffusion. We believe these aspects constitute a substantial theoretical and practical distinction from the original EBGAN framework.
>
> ---

---

### Official Review · Reviewer_4SYN · 2025-10-31

**Soundness:** 4
**Presentation:** 3
**Contribution:** 4
**Rating:** 8
**Confidence:** 4

**Summary:**

This paper proposes Discrepancy-aware Score Learning (DSL), a novel adversarial training framework for diffusion models that integrates an energy-based discriminator with a margin-based hinge loss in the noise space. DSL aims to address the over-smoothing issue in denoising score matching by adaptively reweighting gradients to focus on challenging samples. The authors provide a theoretical interpretation of DSL as functional gradient descent and demonstrate its effectiveness across multiple generative tasks, including text-to-image generation, conditional synthesis, super-resolution, and 2D-to-3D reconstruction.

**Strengths:**

The work introduces a novel and generalizable framework that enhances diffusion training without architectural changes. It offers both theoretical insights and practical improvements, demonstrating consistent gains in fidelity, perceptual quality, and semantic alignment across multiple domains.

**Weaknesses:**

The theoretical section, while rigorous, may be challenging for readers less familiar with functional gradient flows. The method’s performance is sensitive to the adversarial weight λ, though the authors provide a sensitivity analysis. More comparisons with recent adversarial diffusion methods (e.g., ADD, Structure-guided Adv. Training) could further strengthen the claims.

**Questions:**

- How does DSL scale with larger models or datasets beyond LAION-5B?
- Could the margin \( m \) be learned adaptively rather than set via mean error?
- Have the authors considered applying DSL to other modalities (e.g., audio, video) as suggested in the conclusion?

---

> ### Author Response · Authors · 2025-11-21
> **Response to Reviewer 4SYN (1/2)**
>
> Thank you for your insightful feedback. Your comments helped us clarify our key contributions.
>
> >**Q1.  How does DSL scale with larger models or datasets beyond LAION-5B?**
>
> **A1.** Thank you for your thoughtful comment on scalability.
> Recent large-scale models such as SD 3.5 [1] and FLUX.1 [2] often construct training corpora by aggregating multiple web-scale datasets (*i.e.*, ImageNet and CC12M) with additional proprietary filtering. However, the exact composition and filtering pipelines of these mixtures are typically not fully specified or publicly released, which makes faithful reproduction and controlled comparisons difficult.
>
> In contrast, we intentionally conducted our experiments on LAION-5B to ensure that both the dataset and our training setup are reproducible. LAION-5B contains ~5.85B image-text pairs, which is already at the scale of current state of the art text-to-image systems. We believe demonstrating consistent improvements at this scale provides strong evidence that DSL scales to large models and large datasets in practice.
>
> From a computational perspective, DSL does not require any architectural changes to the backbone diffusion model. It only augments the training objective with an energy-based discriminator that reuses a frozen copy of the generator backbone and adds lightweight LoRA adapters, as detailed in Sec. 3.1 and Sec. B.
> As a result, the asymptotic computational and memory scaling with respect to model size and dataset size remains comparable to the underlying diffusion model, with a moderate per-step overhead but faster convergence in terms of training iterations.
>
> To facilitate verification on even larger models and alternative dataset mixtures beyond LAION-5B, we will release our full training code and configuration files. This will allow practitioners to plug DSL into their preferred large-scale pipelines and evaluate its behavior under different data and model scaling regimes.
>
> [1] Esser, P., Kulal, S., Blattmann, A., Entezari, R., et al., 2024. Scaling rectified flow transformers for high-resolution image synthesis. International Conference on Machine Learning.
>
> [2] Batifol, S., Blattmann, A., Boesel, F., Consul, S., et al., 2025. FLUX.1 Kontext: flow matching for in-context image generation and editing in latent space. arXiv preprint arXiv: 2506.15742.
>
> ---
>
> >**Q2. Could the margin ($m$) be learned adaptively rather than set via mean error?**
>
> **A2.** We appreciate your insightful comment on the margin design.
> In principle, the margin $m$ can be treated as an additional learnable scalar that is updated adaptively, for example by alternately updating $m$ and the discriminator parameters according to a dedicated objective for $m$. In such a setup, an explicit evaluation criterion is required to determine what value of $m$ is optimal, that is, a separate loss or regularization term that governs the update of $m$ itself.
>
> In our work, instead of introducing an extra training loop for $m$, we adopted a decision-theoretic criterion based on the empirical mean error between real and generated noise, as specified in Sec. 3.1. In detail, we periodically run inference on the training set (*e.g.*, every five epochs), compute the mean squared discrepancy, and set the margin $m$ according to this mean-error criterion. With this choice, the optimal $m$ is obtained by a linear closed-form solution, so $m$ is determined deterministically from data without any additional training procedure or gradient updates specific to $m$.
>
> Empirically, our experiments show that this mean-error based margin works consistently well across different tasks and settings. Moreover, varying $m$ within a moderate range around the resulting value does not lead to significant performance degradation, which indicates that the method is robust to small changes in the margin.
>
> We fully agree that a more sophisticated or explicitly learned margin schedule might further improve performance in some regimes. We view our mean-error based $m$ as a principled and simple default that avoids extra training instability, and we believe that our released code will make it easy for future work to explore alternative learnable margin strategies.
>
> ---

---

> ### Author Response · Authors · 2025-11-21
> **Response to Reviewer 4SYN (2/2)**
>
> ---
> >**Q3. Have the authors considered applying DSL to other modalities (e.g., audio, video) as suggested in the conclusion?**
>
> **A3.** Thank you for highlighting this direction.
> DSL is formulated in the noise space and only requires access to a diffusion-style noise predictor. Therefore, the framework is directly compatible with diffusion models in other modalities, such as audio and video, as long as they are trained with a denoising score matching or closely related objective. In particular, the discriminator loss and margin-based reweighting are defined at the level of predicted noise, not image-specific architectures.
>
> Due to space and computational constraints, we focused our empirical evaluation on image-based tasks (*i.e.*, text-to-image, conditional generation, super-resolution, and 2D-to-3D). Extending DSL to audio, video, and other modalities is a promising avenue that we explicitly mentioned as future work in the conclusion, and we are actively exploring such extensions.
>
> ---

---

> > ### Comment · Reviewer_4SYN · 2025-11-26
> >
> > Thank you for your thorough and thoughtful responses to my questions. I appreciate the clarifications regarding scalability, margin adaptation, and modality extension, which strengthen the reproducibility and practical relevance of your work.
> >
> > - **On scalability**: Your justification for using LAION-5B as a representative large-scale dataset is well-taken, and the use of LoRA to minimize overhead is a pragmatic design choice. Releasing the code will indeed facilitate further validation on even larger models and datasets.
> >
> > - **On adaptive margin**: Your decision-theoretic approach to setting \( m \) is principled and effective. While a learned margin could offer further gains, your method strikes a good balance between simplicity and performance, and the robustness to small variations in \( m \) is reassuring.
> >
> > - **On multi-modal extension**: It is encouraging to hear that DSL is modality-agnostic in principle, and I look forward to seeing future work applying it to audio, video, or other domains.
> >
> > Overall, your responses have addressed my concerns and reinforced the strength and generality of the proposed method. I commend the authors for a well-executed study and a clear, constructive rebuttal.

---

### Official Review · Reviewer_6YSk · 2025-10-31

**Soundness:** 3
**Presentation:** 3
**Contribution:** 3
**Rating:** 8
**Confidence:** 3

**Summary:**

This work introduces an new method for training diffusion models - Discrepancy-aware Score Learning (DSL) - via modified score matching objective. The key idea is to introduce an additional denoising network - discriminator - and supervise it with an EBGAN-like loss, while training the generator (denoiser) by minimizing discrepancy with the discriminator. Authors provide a formal connection of the proposed method to Wassestein gradient flows - and use it to provide convergence guarantees (in equilibrium sense). Experimental evaluation is conducted on multiple generation tasks and suggests that proposed objective performs favorably compared to basic score matching loss.

.

**Strengths:**

+ Paper is clearly written and is easy to follow.
+ Method seems to be well-motivated and authors also provide interesting formal interpretation of the proposed approach in the context of variational inference.
+ Large-scale aside (2 models instead of 1), the method seems to be simple to implement, and according to authors is not sensitive to hyperparameters (e.g. margin).
+ Quantitative and qualitative results seem convincing - the improvements are consistent across multiple different tasks.

**Weaknesses:**

Method limitations:
- It seems that the method requires keeping another copy of a large diffusion model . Given the general trend towards large number of parameters, this makes proposed method significantly less practical.

Evaluation:
- It would be great to understand how the proposed method performs compared to more recent training paradigms (e.g. flow matching) - which tend to produce improved sample quality over score matching. This should help contextualize the performance with respect to SOTA.
- It is unclear if comparison to the models trained same number of iterations is fully fair - as for the proposed method - one de-facto has two models, and needs to update both of them at every iteration. It is OK but being explicit about this is probably going to help the readers.
- Ablation study on architecture / naive adversarial loss is not provided.

**Questions:**

- Could authors confirm whether the evaluation is conducted by re-training the model from scratch or by fine-tuning off-the-shelf weights?
- Would it be possible to provide some numbers on the memory requirements and consequences to speed of training with additional discriminator in terms of wall clock?

---

> ### Author Response · Authors · 2025-11-21
> **Response to Reviewer 6YSk**
>
> Thank you for your constructive and insightful feedback. We appreciate your detailed comments, which helped us clarify key aspects and improve the presentation of our work.
>
> >**Q1. Could authors confirm whether the evaluation is conducted by re-training the model from scratch or by fine-tuning off-the-shelf weights?**
>
> **A1.** We appreciate the insightful comment.
> For computational efficiency, all experiments in Sec. 3 (text-to-image, conditional generation, super-resolution, and 2D-to-3D) were conducted by fine-tuning publicly available pre-trained checkpoints (*e.g.*, SDXL) rather than training large-scale models from scratch.
>
> To demonstrate that DSL does not rely on off-the-shelf weights and is applicable in a from-scratch training regime, Sec. B and Fig. 7 of the supplemental material present results where a model with the same architecture is trained from scratch. These experiments show that DSL still improves convergence speed and final performance in this setting.
>
> To avoid confusion, we have revised Sec. 3.1 and Sec. B to explicitly state that:
> 1. all main results are obtained by fine-tuning pre-trained models, and
> 2. additional from-scratch experiments with the same architecture are provided in the supplemental material.
>
> In the revised paper, all ***updated parts*** are highlighted in blue for clarity. These color markings are for review purposes only and will be reverted to the original color in the camera-ready version.
>
> ---
>
> >**Q2. Would it be possible to provide some numbers on the memory requirements and consequences to speed of training with additional discriminator in terms of wall clock?**
>
> **A2.** We appreciate this helpful suggestion and agree that clarifying the computational overhead of the discriminator is important.
>
> *Discriminator architecture and memory.*
> We adopted a standard Low-Rank Adaptation (LoRA) [1] for the discriminator. Concretely, the discriminator reuses a frozen copy of the generator backbone. Instead of updating the backbone parameters, we inserted rank-8 learnable low-rank matrices, *i.e.*, LoRA adapters, into the linear projection layers of the attention modules inside the frozen U-Net ($\sim$2.6B) and trained only these additional parameters for the discriminator. This design keeps the number of trainable parameters and activations ($\sim$24M) for backpropagation relatively small. In our SDXL experiments, this did not require any reduction in batch size or change in hardware configuration compared to the baseline; the same GPUs and batch sizes were used for both methods.
>
> *Wall-clock time per step.*
> The discriminator introduces one additional forward pass through the frozen backbone during training. Under the same GPU setting, this increases the wall-clock time per training step by ~19.3%.
>
> *End-to-end training time and total cost.*
> Despite the moderate per-step overhead, DSL converges more quickly. In our SDXL text-to-image experiment, the proposed method reaches an FID of 9.32 after 1.2M training steps, whereas the baseline attains an FID of 13.81 only after 2.0M steps, as reported in Sec. B and Fig. 7 of the supplemental material. This corresponds to ~40% fewer training steps, yielding ~28.4% reduction in total training time, while also improving FID by ~32.5%.
>
> For clarification, we have revised Sec. 3. 1 and Sec. B to explicitly describe the discriminator implementation, the per-step wall-clock overhead, and the convergence behavior relative to the baseline, stating that:
> 1. the additional discriminator can be implemented with modest memory overhead by reusing the frozen generator backbone with lightweight LoRA adapters,
> 2. per-step wall-clock time increases moderately, and
> 3. faster convergence and better final performance reduce the overall computational cost and total training time.
>
> [1] Hu, E.J., Shen, Y., Wallis, P., Allen-Zhu, Z., Li, Y., Wang, S., Wang, L., Chen, W., 2022. LoRA: Low-rank adaptation of large language models. International Conference on Learning Representations.

---

### Meta-Review · Area_Chair_rpsC · 2026-01-13

**Summary:**

While two reviewers found the submission to be technically sound and empirically results to be strong, a serious concern was raised by one of the reviewers regarding novelty and positioning relative to prior work. It was pointed out that the proposed framework is highly similar in methodology, training heuristics, and experimental protocol to an ICCV 2025 paper "Generative Adversarial Diffusion" (GAD). The overlap includes the use of a margin-based adversarial regularizer in the noise space, an identical margin-setting heuristic based on mean error computed every five epochs, similar robustness checks, and a largely overlapping experimental set-up. More importantly, this work is not even cited, which further weakens the contribution claim. Even though I do not wish to dismiss the paper for plagiarism, the high similarly indeed raises serious concerns. Beyond this issue, reviewers also noted missing ablations, limited comparisons to newer diffusion training paradigms, and incomplete empirical validation of some core claims. Due to these reasons, I cannot recommend acceptance.

**Reviewer Concerns:**

Despite the rebuttal's attempt at addressing practical and clarity-related concerns, the key issue regarding the high degree of similarity with the ICCV 2025 paper (not even cited) remained.

**Reviewer Scores:**

While Reviewer 4SYN and Reviewer 6YSk maintained their original scores, they seemed to be unaware of the similarity with the ICCV 2025 paper, becoming familiar with which I suspect their scores to have reduced. More importantly, reviewer Mgsj did not move from their original score (2).

---

### Decision · Program_Chairs · 2026-01-26

Reject